# Tunnel-structured IrO$_x$ unlocks catalytic efficiency in proton exchange membrane water electrolyzers

Mingcheng Zhang[1], Wei An[1], Qianqian Liu[1,2], Yuzhu Jiang[1], Xiao Zhao [3], Hui Chen [1], Yongcun Zou[1], Xiao Liang [1] ✉ & Xiaoxin Zou [1] ✉

Proton exchange membrane water electrolyzers face challenges due to high iridium loading and sluggish oxygen evolution reaction kinetics when using conventional rutile-structured iridium oxide nanocatalysts. Here we find that iridium oxide catalysts with a specific tunnel-type crystal structure exhibit highly localized reactivity, where regions at tunnel mouths drive oxygen evolution far more efficiently than tunnel-wall regions. The intrinsic activity of tunnel mouths is 25-fold higher than that of tunnel walls, with shorter nanorods achieving a better balance between active site exposure and electron/mass transport efficiency. When implemented in proton exchange membrane water electrolyzers, this engineered catalyst achieves notable performance at low iridium loading (0.28 mg$_{Ir}$ cm$^{-2}$), delivering over 2.0 A cm$^{-2}$ at 1.8 V (80 °C) and operating stably for 1800 h—notably outperforming conventional catalysts. Our work identifies catalytic hotspots in tunnel-structured oxides and demonstrates their rational integration into high-performance, durable electrolyzer systems.

As the only electrolysis technology capable of coupling gigawatt-scale hydrogen production with millisecond-scale dynamic response, proton exchange membrane water electrolyzers (PEMWEs) have become the critical enabler for converting intermittent renewable power into industrial-grade green hydrogen[1–3]. With PEMWE installations expected to reach 80–100 GW by 2030, a severe bottleneck threatens scalability: projected iridium demand (1.5–2.0 tons/year) risks exhausting ~30% of global supply[4–6]. This stems from PEMWE's exclusive reliance on iridium-based catalysts, where conventional rutile-phase IrO$_x$ nanoparticles demand excessive loadings of 2–4 mg$_{Ir}$ cm$^{-2}$ to sustain industrial activity and durability standards at ampere-level current densities[7–9]. The crisis intensifies when contextualizing iridium's scarcity. Its annual production (~7 tons) and crustal abundance (0.001 ppm) are lower than those of platinum (~200 tons; 0.005 ppm), while its price (>$6500/oz) is much higher than platinum's ($1000/oz)[10–12]. To sustain PEMWE's exponential growth, iridium utilization

efficiency must improve substantially, without compromising operational activity and stability[13,14].

Enhancing iridium utilization in PEMWE electrocatalyst design faces a critical dilemma: reducing Ir loading inherently compromises the catalyst layer's triple requirements of active site abundance, electrical connectivity, and mechanical robustness[15–17]. In conventional rutile-phase IrO$_x$ nanoparticle systems, decreasing loadings below 0.5 mg$_{Ir}$ cm$^{-2}$ often disrupts the catalyst layer's structural integrity (Fig. 1a), inducing an exponential rise in electrically isolated catalyst clusters (islanding effect)[18,19]. This typically leads to three interrelated failure mechanisms[20,21]. (1) Insufficient active sites limit high-current operation; (2) Disrupted conductive networks elevate in-plane resistance and interfacial contact losses; (3) Uncontrolled particle migration intensifies under high current densities (>1.0 A cm$^{-2}$), where combined electric field gradients and gas-liquid shear forces propel nanoparticle agglomeration, further exacerbating islanding effect and

[1]State Key Laboratory of Inorganic Synthesis and Preparative Chemistry, College of Chemistry, Jilin University, Changchun, China. [2]School of Materials Science and Engineering, Xi'an University of Science and Technology, Xi'an, China. [3]Key Laboratory of Automobile Materials of MOE, School of Materials Science and Engineering, Jilin Univiersity, Changchun, China. ✉e-mail: liangxiao@jlu.edu.cn; xxzou@jlu.edu.cn

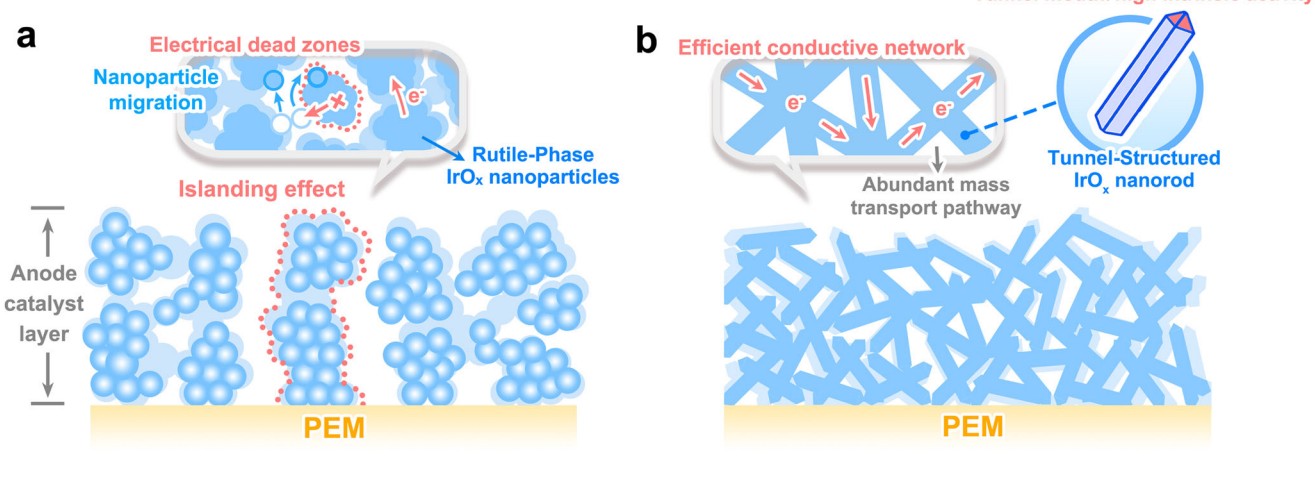

**Fig. 1 | Comparison of conventional and hierarchical anode assemblies.**
**a** Schematic of conventional anode assembly based on rutile-phase IrO$_x$ nanoparticles in a proton exchange membrane (PEM) electrolyzer, illustrating degradation mechanisms such as nanoparticle migration and islanding effect, where discontinuous catalyst domains lead to electrical dead zones. **b** Schematic of hierarchical anode assembly based on tunnel-structured IrO$_x$ nanorods, featuring an efficient electron-conductive network and abundant mass transport pathways.

activity decay. Simultaneously, high current operations also cause severe oxygen bubble coalescence and high diffusion-limited overpotentials—a persistent bottleneck responsible (20–30%) for efficiency loss in PEMWE[22,23]. These intertwined challenges require multiscale structural optimizations that combine atomic-scale catalyst tuning to maximize intrinsic activity, stability, and 3D hierarchical electrode architectures engineered with robust conductive skeletons and mass transport channels. This integrated approach could decouple reaction kinetics from iridium loading while ensuring rapid electron/gas/liquid transport across scales.

This objective motivates our exploration of tunnel-structured iridium oxides (T-IrO$_x$), which naturally exhibit unique 1D nanostructures (nanofibers/wires) enabling self-assembled conductive networks and mass transfer channels[24,25]. Besides, the iridium oxidation state in T-IrO$_x$ is intrinsically stabilized at sub-4 through charge compensation by interstitial H$^+$ species, which is beneficial for resisting over-oxidation and maintaining the structural integrity of the framework during harsh OER operations[26]. Paradoxically, despite their structural advantages, T-IrO$_x$ nanocatalysts underperform conventional R-IrO$_x$ nanoparticles in PEMWEs, even at high iridium loadings (>1 mg$_{Ir}$ cm$^{-2}$)[25,27–29]. In this work, we resolve this contradiction by revealing intrinsic activity heterogeneity within T-IrO$_x$—an aspect previously overlooked. Through combined DFT and operando characterization, we demonstrate that tunnel mouths (termini of 1D channels) exhibit 25 times higher oxygen evolution activity than tunnel walls, providing valuable insights into the design of iridium oxide catalysts. By strategically synthesizing short T-IrO$_x$ nanorods to increase tunnel-mouth exposure while preserving electron/mass transport pathways (Fig. 1b), we achieve promising PEMWE performance at 0.28 mg$_{Ir}$ cm$^{-2}$: >2.0 A cm$^{-2}$@1.8 V with 1800 h durability. This work highlights the critical role of nanoscale active site localization in designing high-performance electrocatalysts for sustainable hydrogen production.

## Results

### Theoretical activity and electronic structure calculation
Tunnel-structured IrO$_x$ nanomaterials can be derived from tunnel-structured iridates via acid leaching of alkali metals in them. Tunnel-structured iridates are a subgroup of microporous transition metal

oxides (also known as octahedral molecular sieves), which are characterized by 1D tunnels constructed with regular walls of edge-shared oxygen octahedra[30–32]. A representative tunnel iridate with a 2 × 2 tunnel configuration is selected to describe the crystal chemistry of tunnel iridates. As illustrated in Fig. 2a, IrO$_6$ octahedra form edge-sharing chains along the $c$-axis, and two chains further interlock like a zipper to create octahedral ribbons (Fig. 2a, inset). Four ribbons are orthogonally connected in a corner-shared manner to form the tunnel structure, whose cavity is typically occupied by K ions. The structural diversity of tunnel configurations in tunnel iridates family is due to the variation in width and connectivity of octahedral ribbons (i.e., tunnel walls). As shown in Fig. 2b, the width of the ribbons can range from one to three units, and they can be connected in either orthogonal or parallel modes. To explore the potential impact of these structural differences on catalytic activity, another four tunnel iridates with different tunnel configurations are also discussed in this work. For simplicity, these tunnel iridates are labeled according to their dimensions of tunnel mouths (Fig. 2c). To clarify, only the 2 × 2 tunnel configuration has been experimentally accessible to date, the other tunnel oxide configurations have not yet been reported to be synthesized and are considered theoretical prototypes in this work.

Considering that tunnel iridates serve as anode catalysts in PEMWE under certain pH conditions (pH-2), the impact of pH on their crystal structures is investigated by taking the tunnel iridate with a 2 × 2 tunnel configuration as an example. Figure 3a shows its Gibbs free energy of K$^+$ leaching ($\Delta G_{leaching}$) at varying pH values. The negative $\Delta G_{leaching}$ values observed at pH below 9 indicate that K$^+$ is inclined to be spontaneously replaced by H$^+$ (i.e., K$^+$/H$^+$ exchange) in such a situation. The consistent $\Delta G_{leaching}$ vs. pH trends are also observed for the other four tunnel-structured iridates (Supplementary Fig. 1). In addition, we evaluated the accessibility of the tunnel interior to water molecules. For the 1 × 2 and 1 × 2′ tunnels, the small tunnel dimension and significant steric hindrance prevent water entry. Larger tunnels (2 × 2, 2 × 3, and 3 × 3) can accommodate water molecules, but the thermodynamic driving force for water insertion is small (0.02–0.03 eV/atom; Supplementary Table 1). These results imply that these tunnel iridates ultimately exist as protonated iridium oxide with a tunnel-structured framework in the operating circumstance of PEMWE.

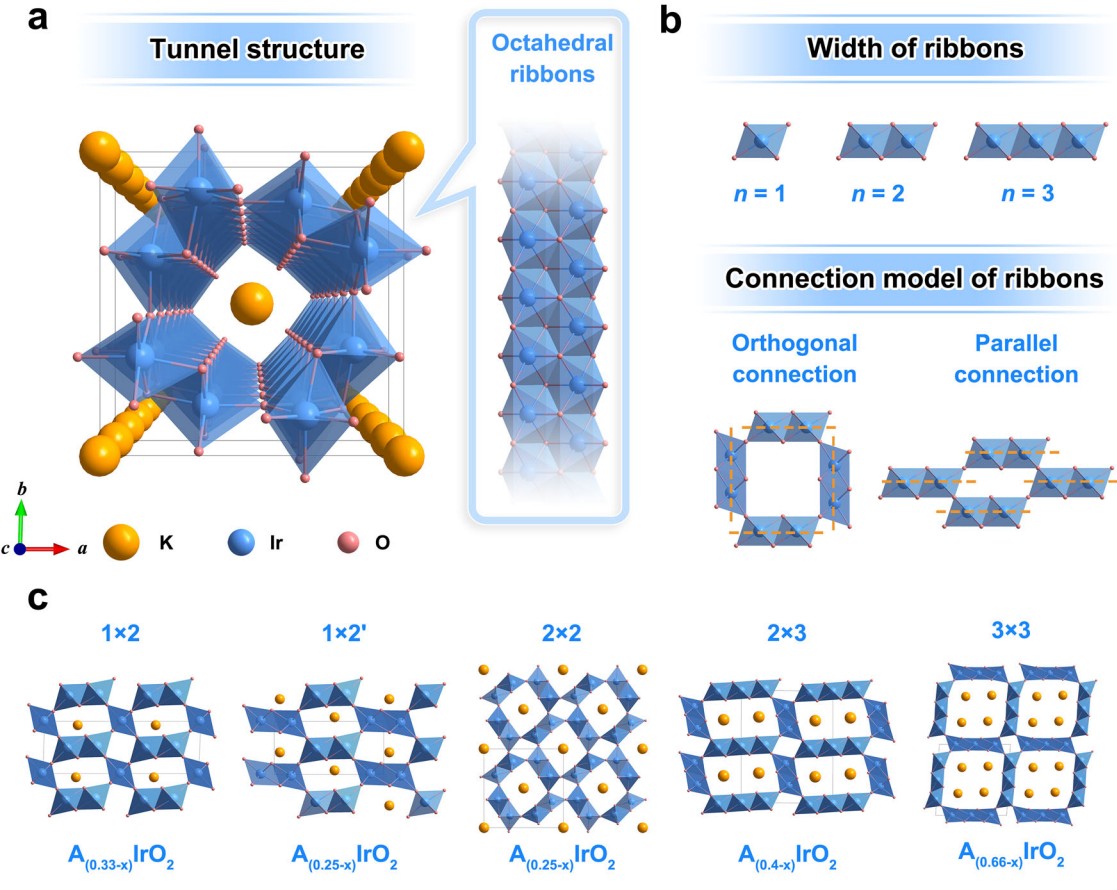

**Fig. 2 | Structural diversity of tunnel-structured iridates. a** Crystal structure of a representative tunnel-structured iridate with a 2 × 2 tunnel configuration. The inset highlights the octahedral ribbons that form the tunnel structure. **b** Schematic for various widths and connection models of octahedral ribbons. **c** Crystal structures of five tunnel-structured iridates, annotated with their chemical formulas and corresponding tunnel types. The tunnel cavities are occupied by alkali metal ions. Note: The tunnel mouths of the 1 × 2 and 1 × 2′ samples possess comparable dimensions, but with distinct structural configurations. The 2 × 2 tunnel-structured iridate corresponds to the structure shown in Fig. 2a.

We next study the stable surface configurations at different applied potentials by constructing phase diagrams of all the low-index surfaces of the five tunnel iridium oxides[33,34]. Figure 3b shows that Ir sites on the (001) surface of 2 × 2 tunnel iridium oxide are preferentially occupied by oxygen in the potential range for OER (i.e., >1.6 V vs. RHE). Actually, similar results are observed for all the low-index surfaces of the five tunnel iridium oxides (Supplementary Figs. 2–6). Thus, the theoretical models with O*-covered surfaces are selected for subsequent calculations on catalytic activities and electronic structures.

The calculation of OER activity is based on the four-electron transfer pathway (i.e., the adsorbate evolution mechanism, AEM) as proposed by Nørskov et al.[33,35]. The possibility of OER occurring at the inner active sites are evaluated given that water molecules can enter the interiors of 2 × 2, 2 × 3, and 3 × 3 tunnel iridium oxides. Most (82%) of these inner sites are unable to stabilize key OER intermediates OOH*, while the remaining (18%) exhibit overpotentials exceeding 1.5 V, indicating that these inner active sites are inert (Supplementary Table 2). Therefore, the catalytic activity is predominantly determined by the external surfaces of the tunnel iridium oxides.

In Fig. 3c, the theoretical OER overpotentials for all the low-index surfaces of the five tunnel iridium oxides are presented. The different tunnel types are distinguished by various shapes, with blue and green representing the surfaces related to tunnel mouths (characterized by the (hk1) index, hereafter referred to as tunnel mouths) and tunnel walls (characterized by the (hk0) index, hereafter referred to as tunnel walls), respectively. To better visualize their adsorption properties and theoretical activities, the corresponding data points are plotted on an activity volcano plot, which is constructed by scaling relationships for the adsorption Gibbs free energies of intermediates (i.e., $\Delta G_{OH}$, $\Delta G_O$, and $\Delta G_{OOH}$) (Supplementary Fig. 7)[36,37]. For comparison, the theoretical activity of rutile $IrO_2$ (110) surface is also indicated in the Fig. 3c as a grey inverted triangle.

As shown in Fig. 3c, regardless of tunnel configurations, the tunnel mouths are closer to the volcano peak than the corresponding tunnel walls, suggesting the higher catalytic activity of the former. For instance, the (001), (101), and (111) surfaces of 2 × 2 tunnel iridium oxide (i.e., three tunnel mouth-related surfaces) show good OER activities with theoretical overpotentials of 0.38 V, 0.43 V, and 0.50 V, respectively (Supplementary Table 3). In contrast, the (100) and (110) surfaces of 2 × 2 tunnel iridium oxide (i.e., two tunnel wall-related surfaces) show theoretical overpotentials of 0.56 and 0.72 V, demonstrating the catalytic activity of tunnel walls is similar to the (110) surface of rutile $IrO_2$. Additionally, the catalytic activity of tunnel mouths across various tunnel configurations remains at a similar level. These results suggest that the higher catalytic activity of tunnel mouths and the nonuniform distribution of catalytic activity on catalyst particles could be an intrinsic property of tunnel iridium oxides. To maximize catalytic performance of tunnel iridium oxides, it is therefore desirable to increase the exposure of tunnel mouth sites as much as possible.

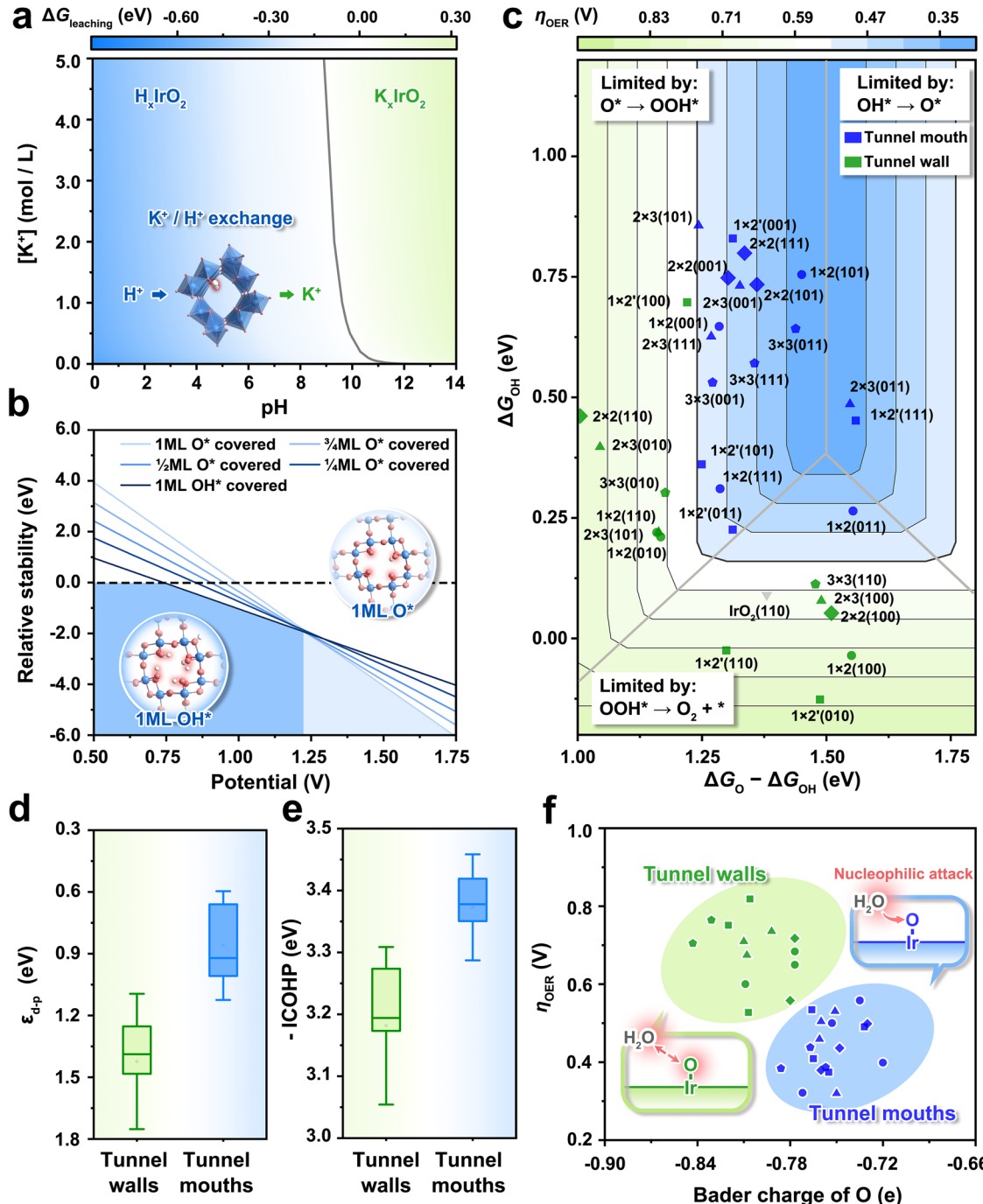

**Fig. 3 | Theoretical analysis of T-IrO$_x$ for OER electrocatalysis. a** The $\Delta G_{leaching}$ of K$^+$/H$^+$ exchange in a tunnel iridate with a 2 × 2 tunnel configuration at different pH. **b** The surface phase diagrams for the (001) surface of 2 × 2 tunnel iridium oxide. **c** Theoretical activity ($\eta_{OER}$) volcano plot using O* and OH* binding energies ($\Delta G_{OH}$ and $\Delta G_O - \Delta G_{OH}$) as descriptors. Point shapes indicate tunnel types. Box plot of **d** charge transfer energies ($\varepsilon_{d-p}$) and **e** integrated crystal orbital hamilton populations (−ICOHP) of Ir-O bonds of tunnel walls and mouths in these tunnel iridium oxides. Boxes show interquartile ranges (25th–75th percentiles), whiskers show standard deviations, and medians are marked by center lines. **f** Bader charge of oxygen in these tunnel iridium oxides. The inset illustrates the nucleophilic attack of H$_2$O on O*. Source data are provided as a Source data file.

To further investigate the origin for high activity of tunnel mouths, we calculated the electronic structures of these tunnel iridium oxides. Figure 3d, Supplementary Figs. 8, 9 and Table 4 demonstrate that the charge transfer energies ($\varepsilon_{d-p}$) at tunnel mouths (0.66–1.05 eV) are considerably lower than those at tunnel walls (1.15–1.65 eV), suggesting a stronger overlap between Ir d-band and O p-band at tunnel mouths[38,39]. This is further supported by the stronger Ir-O covalency at the tunnel mouths, as quantified by the integrated

Crystal Orbital Hamilton Populations (ICOHP, Fig. 3e and Supplementary Fig. 10). The enhanced Ir-O bond covalency of tunnel mouths modulates the excessively strong adsorption energy of intermediates, potentially contributing to their higher theoretical catalytic activity[38]. On the other hand, Fig. 3f presents Bader charge analysis, which shows that O on the tunnel mouths possess a more positive charge range (from −0.78 to −0.72 e$^-$) than those on the tunnel walls (from −0.84 to −0.78 e$^-$). The negative charge on O* on the tunnel mouths is more

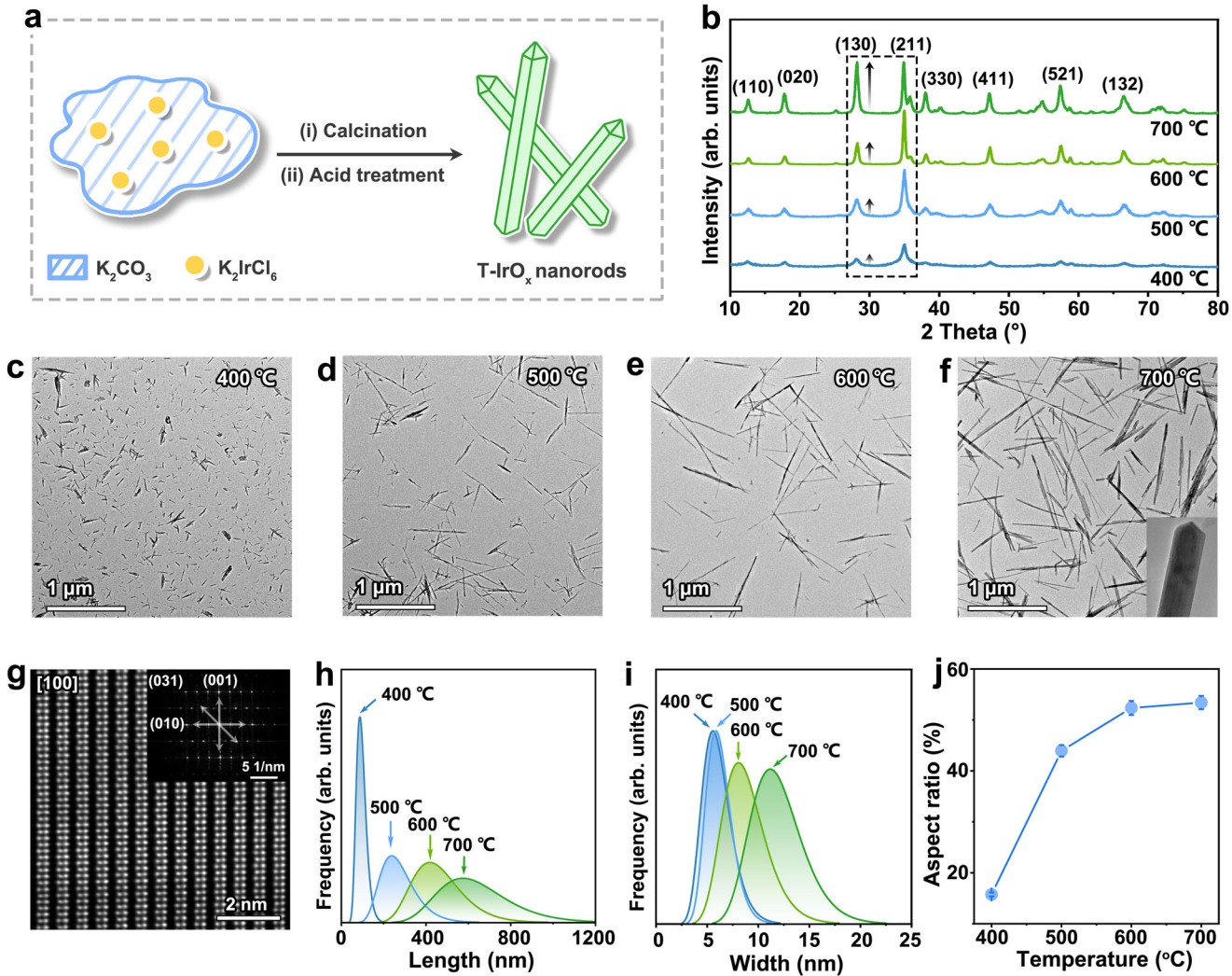

**Fig. 4 | Synthesis and structural characterization of T-IrOₓ nanorods.**
**a** Schematic of the synthesis process for T-IrOₓ nanorods. **b** XRD patterns of T-IrOₓ synthesized at different temperatures. TEM images of T-IrOₓ synthesized at **c** 400 °C, **d** 500 °C, **e** 600 °C, and **f** 700 °C, with the inset in **f** showing a detailed view of the T-IrOₓ-700 nanorod tip. **g** HAADF-STEM image and corresponding FFT pattern of T-IrOₓ-400. Statistical distribution of **h** length and **i** width of T-IrOₓ nanorods synthesized at different temperatures based on 200 randomly selected rods per condition. **j** Aspect ratio of T-IrOₓ nanorods versus synthesis temperature. Error bars indicate standard deviation. Source data are provided as a Source data file.

susceptible to nucleophilic attack by H₂O, thereby accelerating the OER kinetics.

## Synthesis and structural characterization of T-IrOₓ

To validate the nonuniform activity distribution predicted by theoretical calculations, we experimentally synthesized 2 × 2 tunnel iridium oxide (referred to as T-IrOₓ hereafter) nanorods with different aspect ratios. The synthesis is realized by a low-temperature solid-state reaction of K₂CO₃ and K₂IrCl₆ at a broad temperature range from 400 to 700 °C, followed by acid treatment at room temperature (Fig. 4a, see detailed synthesis procedure in the Methods). At temperatures below 400 °C, the reaction yields amorphous products, as confirmed by XRD analysis (Supplementary Fig. 11). While previous reports have developed methods for synthesizing tunnel iridates, these works typically involve multiple reagents and complex steps, such as hydrothermal or sol-gel treatments prior to calcination[28,29]. Moreover, the calcination temperature usually exceeds 600 °C, resulting in the synthesis of long nanofibers of tunnel iridates.

Figure 4b presents the X-ray diffraction (XRD) patterns of T-IrOₓ samples synthesized at 400, 500, 600, and 700 °C, which are denoted as T-IrOₓ-400, T-IrOₓ-500, T-IrOₓ-600, and T-IrOₓ-700, respectively. These patterns demonstrate that the materials share the identical crystal structure. Rietveld refinement of the XRD data for T-IrOₓ-400 matches the crystal parameters of tunnel iridium oxide, confirming that these samples are in the pure phase (Supplementary Fig. 12 and Table 5). With increasing synthesis temperatures, the diffraction peak corresponding to the tunnel wall (e.g., (130) and (330) planes) intensifies, reflecting the aspect ratio evolution of T-IrOₓ nanorods. This conclusion is confirmed by transmission electron microscopy (TEM) images (Fig. 4c–f), which illustrate the nanorod morphology of all samples and a clear lengthening trend of nanorods with rising temperatures. X-ray absorption near-edge structure (XANES) analysis of the Ir L₃-edge (Supplementary Fig. 13) shows the Ir oxidation state of 3.2 ± 0.2 in T-IrOₓ, which is consistent with the sub-4 oxidation state characteristic of tunnel iridium oxides. Extended X-ray absorption fine structure (EXAFS) analysis (Supplementary Fig. 14 and Table 6) identifies three main coordination shells: Ir-O, Ir-Ir, and Ir-Ir' at 2.0 Å, 3.1 Å, 3.5 Å, respectively, in good agreement with the local atomic structure of the tunnel framework.

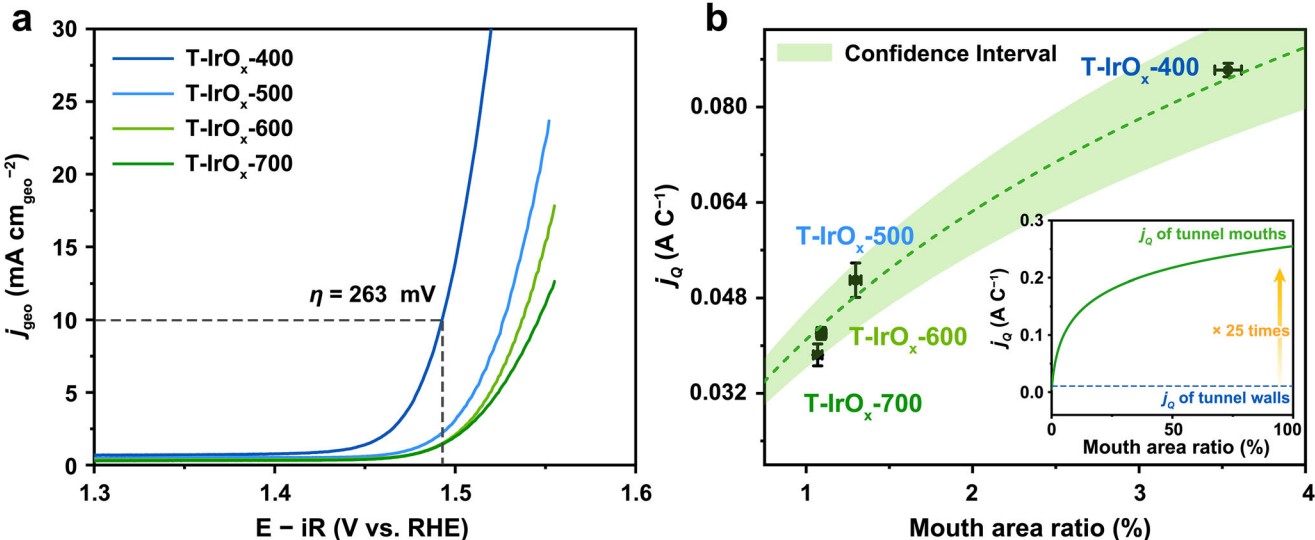

**Fig. 5 | Experimental validations of high intrinsic activity at tunnel mouth. a** iR-corrected OER polarization curves of T-IrO$_x$-400, T-IrO$_x$-500, T-IrO$_x$-600, and T-IrO$_x$-700 measured in 0.1 M HClO$_4$ electrolyte. The measurements were performed in O$_2$-saturated 0.1 M HClO$_4$ (pH 1) at 25 °C, with a scan rate of 1 mV s$^{-1}$, the catalyst loading of 0.281 mg cm$^{-2}$, and O$_2$ flow rate of 10 mL min$^{-1}$. E–iR presents iR-corrected potentials. $j_{geo}$ presents current density which is normalized to geometric electrode area. The compensated resistances, measured via the iR compensation function of the electrochemical workstation, were 35.3 ± 1.2, 35.7 ± 1.1, 31.2 ± 0.1, and 35.2 ± 0.4 Ω for T-IrO$_x$-400, T-IrO$_x$-500, T-IrO$_x$-600, and T-IrO$_x$-700,

respectively. **b** Intrinsic activities ($j_Q$) at 1.54 V vs. RHE of T-IrO$_x$ synthesized at different temperatures versus their mouth area ratio. Intrinsic activity is calculated by normalizing current density by pseudocapacitive charge ($Q$). Mouth area ratio is the ratio of bipyramidal area and surface area of nanorod. The error bars of mouth area ratio and $j_Q$ represent the standard errors from two hundred and three measurements, respectively. Inset shows the fitting logarithmic curve of the relationship between $j_Q$ and mouth area ratio. Source data are provided as a Source data file.

High-magnification TEM images for nanorod tips of T-IrO$_x$ series (Fig. 4f inset and Supplementary Figs. 15–18) reveal that their tip geometries vary systematically with synthesis temperature. Higher-temperature samples (e.g., T-IrO$_x$-700, Fig. 4f inset and Supplementary Fig. 18) exhibit well-developed prismatic and bipyramidal forms, resulting in sharper and more faceted terminations due to more complete nanorod growth. In contrast, lower-temperature samples (e.g., T-IrO$_x$-400, Supplementary Fig. 15) display comparatively smoother and slightly rounded nanorod tips, indicative of less complete growth. Figure 4g and inset show the high-angle annular dark-field scanning transmission electron microscopy (HAADF-STEM) and corresponding fast Fourier transform (FFT) images of prismatic form. The observed lattice fringes, with spacings of 0.50 and 0.16 nm, are assigned to the (002) and (020) planes. These results confirm that the prismatic crystal faces are aligned with the tunnel wall (100) plane, and the crystal elongates along the [001] direction. DFT calculations further explain this anisotropic growth behavior, revealing that the (100) plane has the lowest surface free energy (1.07 J m$^{-2}$), notably lower than the tunnel mouth surfaces, such as the (111) plane (1.55 J m$^{-2}$) (Supplementary Fig. 19 and Table 7). As a result, the tunnel wall surfaces are more favorably exposed compared to the tunnel mouth surfaces, leading to the growth of the 1D nanostructures.

To determine the aspect ratios of T-IrO$_x$ nanorods synthesized at different temperatures, we measured the length and width by randomly analyzing two hundred nanorods from TEM images of each sample (Figs. 4c–f and Supplementary Figs. 20–23). As shown in Fig. 4h and Supplementary Fig. 24, the nanorod length increases with synthesis temperature, rising from 96 nm for T-IrO$_x$-400 to 274, 463, and 640 nm for T-IrO$_x$-500, T-IrO$_x$-600, and T-IrO$_x$-700, respectively. In contrast, the nanorods width shows less temperature dependence, increasing slightly from 6 nm for T-IrO$_x$-400 to 12 nm for T-IrO$_x$-700 (Fig. 4i). Consequently, a significant increase in the aspect ratio of T-IrO$_x$ nanorods with temperature is observed in Fig. 4j, resulting from the pronounced difference in the growth rates of length and width. The effect of calcination time is examined at a fixed temperature of

400 °C. Calcination for less than 2 h results in the formation of amorphous IrO$_x$, while extending the duration beyond 2 h leads to only a slight increase in nanorod length. These results indicate that calcination time has limited influence on the size of nanorods (Supplementary Figs. 25–27).

**Activity and stability evaluation in three-electrode system**
Subsequently, OER activities of the four T-IrO$_x$ samples are evaluated in a three-electrode system using iR-corrected linear sweep voltammetry (LSV). As shown in Fig. 5a, the apparent activity ($j_{geo}$, current density normalized by geometric area of working electrode) decreases from T-IrO$_x$-400 to T-IrO$_x$-700. The overpotentials at $j_{geo}$ = 10 mA cm$^{-2}$ are ~263, 282, 298, and 312 mV for T-IrO$_x$-400, T-IrO$_x$-500, T-IrO$_x$-600, and T-IrO$_x$-700, respectively. For comparison, catalytic activities are collected for rutile IrOx synthesized via the Adams method, nano-crystalline rutile IrO$_2$ from representative literature[40], and several commercial rutile IrO$_x$ nanoparticles. As shown in Supplementary Fig. 28, T-IrO$_x$-400 outperforms these rutile IrO$_x$ benchmarks, whose overpotentials are in the range of 295–333 mV. Additionally, the catalytic activity of T-IrO$_x$-400 ranks among the top for reported iridium-based electrocatalysts (Supplementary Table 8), including perovskite-type (e.g., SrIrO$_3$)[41], pyrochlore-type (e.g., Y$_2$Ir$_2$O$_7$)[42], layered (e.g., Ca$_2$IrO$_4$) iridates[43], and doped iridium oxide (e.g., W$_{1-x}$Ir$_x$O$_{3-\delta}$)[44].

Tafel slope analysis was performed to investigate the morphology-dependent OER kinetics. As shown in Supplementary Fig. 29, the Tafel slope increases with nanorod aspect ratio (T-IrO$_x$-400: 44 mV dec$^{-1}$, T-IrO$_x$-500: 48 mV dec$^{-1}$, T-IrO$_x$-600: 53 mV dec$^{-1}$, T-IrO$_x$-700: 57 mV dec$^{-1}$), suggesting that samples with lower aspect ratios exhibit faster OER kinetics, which is consistent with LSV results. Furthermore, the Tafel slope provides insight into the rate-determining step (RDS) of OER. In T-IrO$_x$ series, the measured Tafel slopes reflect combined contributions from both tunnel-wall and tunnel-mouth sites. For longer nanorods with fewer exposed mouth region (e.g., T-IrO$_x$-700), the slope exhibits close to 60 mV dec$^{-1}$, suggesting that OH adsorption and reorganization (Ir + H$_2$O → Ir-OH$_{ads}$* +

$H^+ + e^-$; Ir-OH$_{ads}$* → Ir-OH$_{ads}$) dominate the OER kinetics. As the nanorods shorten and tunnel-mouth exposure increases, the Tafel slope gradually decreases toward 40 mV dec$^{-1}$, indicating that deprotonation of OH intermediate (M-OH$_{ads}$ → M-O$_{ads}$ + H$^+$ + e$^-$) becomes rate-limiting[45–47]. These results highlight the kinetic distinction between tunnel-mouth and tunnel-wall active sites.

To evaluate the intrinsic activity of the four T-IrO$_x$ samples, catalytic currents are first normalized to pseudocapacitive charge ($Q$) to obtain $j_Q$ (Supplementary Fig. 30). The resulting $j_Q$ value at 1.54 V (vs. RHE, reversible hydrogen electrode) exhibits a consistent trend with the apparent activities: T-IrO$_x$-400 > T-IrO$_x$-500 > T-IrO$_x$-600 > T-IrO$_x$-700. To further validate this trend, we determined the electrochemically active surface area (ECSA) using the double-layer capacitance ($C_{dl}$) method and calculated the corresponding ECSA-normalized current density ($j_{ECSA}$) (Supplementary Fig. 31)[48–50]. The $j_{ECSA}$ values at 1.54 V follow the same order as $j_Q$, confirming that the enhanced activity observed for T-IrO$_x$ synthesized at lower temperatures arises from improvements in intrinsic catalytic activity rather than merely increased surface accessibility. This difference in intrinsic activities among the same material system can be attributed to the nonuniform activity distribution of T-IrO$_x$, as predicted by our theoretical calculations.

Guided by the theoretical findings that tunnel mouths possess higher catalytic activity, the distribution of active sites is quantified by calculating the mouth area ratio and explored its correlation with intrinsic activity (see Supplementary Fig. 32 for calculation details). Figure 5b shows a positive correlation between $j_Q$ and the mouth area ratio, where electrocatalysts with higher mouth area ratios display enhanced intrinsic activity. Specifically, T-IrO$_x$-400, with the higher mouth area ratio (3.54%) among all samples, exhibits optimal intrinsic activity. Further logarithmic curve fitting and extrapolation of the relationship between $j_Q$ and mouth area ratio reveal that the intrinsic activity of the tunnel mouths is ~25 times greater than that of the tunnel walls (Fig. 5b, inset). The nonuniform activity distribution of T-IrO$_x$ is therefore identified as the key factor influencing its catalytic performance. This result also explains why the unsatisfied activity of previously studied tunnel iridates: 1D growth habit typically yields long nanowires or nanofibers with extremely low mouth area ratios, severely restricting the exposure of highly active tunnel mouth sites and ultimately leading to inferior catalytic performance.

In addition to aspect ratio, structural defects could potentially contribute to the catalytic performance[51,52], particularly at lower synthesis temperatures. To investigate this possibility, we perform a series of additional characterizations. Electron paramagnetic resonance (EPR) spectroscopy of T-IrO$_x$ shows no signals attributable to unpaired electrons (Supplementary Fig. 33). Furthermore, EXAFS fitting analysis reveals that the Ir-O coordination numbers in T-IrO$_x$ synthesized at different temperatures are close to 6 (Supplementary Table 6), consistent with a fully coordinated structure. These results collectively indicate the absence of oxygen defects in T-IrO$_x$. Regarding possible iridium defects, aberration-corrected TEM images in Fig. 4g display a well-ordered arrangement of iridium atoms with no observable iridium vacancies. Altogether, these findings demonstrate that structural defects are not a significant factor underlying the property differences among T-IrO$_x$ synthesized at different temperatures.

To evaluate their catalytic stability, we performed chronopotentiometric measurements on T-IrO$_x$ synthesized at different temperatures (Supplementary Fig. 34). The results show that among these samples, T-IrO$_x$-400 exhibits the higher performance, maintaining high catalytic activity for nearly 1000 h with only a minimal voltage increase of 3.0 μV h$^{-1}$. T-IrO$_x$-500 and T-IrO$_x$-600 exhibit comparable stability to T-IrO$_x$-400, whereas T-IrO$_x$-700 shows markedly inferior stability, with rapid voltage rise during operation (283 μV h$^{-1}$), which is primarily due to mechanical detachment of oversized nanorods from the electrode substrate. The Faradaic efficiency for OER catalysis is further measured to be 100% (Supplementary Fig. 35), suggesting the complete conversion of current into O$_2$ in the presence of T-IrO$_x$-400 as the catalyst.

To assess the structural stability, inductively coupled plasma atomic emission spectroscopy (ICP-OES) is used to monitor Ir dissolution during OER. As shown in Supplementary Fig. 36, Ir leaching from T-IrO$_x$-400 occurred only during the first few hours, with a leaching content of 0.008 wt%, which is an order of magnitude lower than that of R-IrO$_x$ (0.086 wt%). Such notable structural stability could be attributed to the stabilized sub-4 iridium oxidation state within tunnel framework, as evidenced by XANES analysis (Supplementary Fig. 37) showing an iridium oxidation state of ~3.1 in T-IrO$_x$-400 after OER (T-IrO$_x$-OER). This intrinsic electronic stabilization effectively suppresses iridium over-oxidation—a key degradation pathway for iridium oxides under acidic OER conditions.

Rietveld-refined XRD patterns (Supplementary Fig. 38 and Table 5), Raman spectra (Supplementary Fig. 39) and EXAFS analyses (Supplementary Fig. 40 and Table 6) of as T-IrO$_x$-OER show minimal changes compared to the pristine sample, indicating the structural integrity. The HRTEM image of T-IrO$_x$-OER (Supplementary Fig. 41) further confirms that the nanorod morphology and high crystallinity of T-IrO$_x$-400 are intact, with no evidence of surface amorphization. Similar morphological and crystallographic preservation is also observed for T-IrO$_x$-500, -600, and -700 after OER (Supplementary Figs. 42–44), indicating consistent structural robustness across the T-IrO$_x$ series.

To determine whether lattice oxygen is involved during OER catalyzed by T-IrO$_x$, we employed in situ differential electrochemical mass spectrometry (DEMS) with $^{18}$O isotope labeling. As shown in Supplementary Fig. 45, the $^{34}$O$_2$/$^{32}$O$_2$ ratio in the O$_2$ products from T-IrO$_x$-400 is 0.42%, which is consistent with the natural isotopic abundance of $^{18}$O in water (≈0.21%), suggesting negligible involvement of lattice oxygen during OER. This observation aligns with DFT predictions (Supplementary Fig. 46 and Table 9), further corroborating the catalyst's high structural robustness.

## Performance of T-IrO$_x$-based PEMWE

The catalytic performances at ampere-level current densities of four T-IrO$_x$ samples as anode catalysts are further evaluated in single-cell PEMWEs (see PEMWE configuration in Fig. 6a). Catalyst-coated membranes (CCMs) are fabricated by ultrasonic spray deposition of T-IrO$_x$ and 40 wt% Pt/C onto the anode and cathode sides of a Nafion® 115 membrane. X-ray fluorescence (XRF) spectroscopy confirms that the Ir and Pt loadings are ~0.28 mg cm$^{-2}$ and 0.21 mg cm$^{-2}$ (Supplementary Table 10), respectively. A R-IrO$_x$-based CCM, with similar Ir and Pt loadings, was also prepared for comparative purposes. SEM analysis of the CCM structures revealed the nanorod morphology of the four T-IrO$_x$ samples was retained on the anode side, where the nanorods interweaved with each other to form a continuous catalytic layer (Supplementary Fig. 47). The R-IrO$_x$ nanoparticles form a uniform catalyst layer with the aggregated structure (Supplementary Fig. 48).

The catalytic performances of the CCMs are assessed in single-cell PEMWEs operated at 80 °C and ambient pressure. Figure 6b presents the steady-state polarization curves, which show that the differences in catalytic performance among the four T-IrO$_x$-based PEMWEs align with the activity trend observed in the three-electrode measurements (Fig. 5a and Supplementary Figs. 30–31). The activity trend is as follows: T-IrO$_x$-400 > T-IrO$_x$-500 > T-IrO$_x$-600 > T-IrO$_x$-700. Among these, the T-IrO$_x$-400-based PEMWE gives a current density of 2.0 A cm$^{-2}$ at 1.77 V, whereas the R-IrO$_x$-based PEMWE delivers the same current density at 1.84 V. Furthermore, the T-IrO$_x$-400-based PEMWE outperforms the vast majority of reported PEMWEs with iridium-based catalysts (e.g., Ir black, IrO$_x$, and supported catalysts) in both platinum group metal (PGM) loading and catalytic performance, as summarized

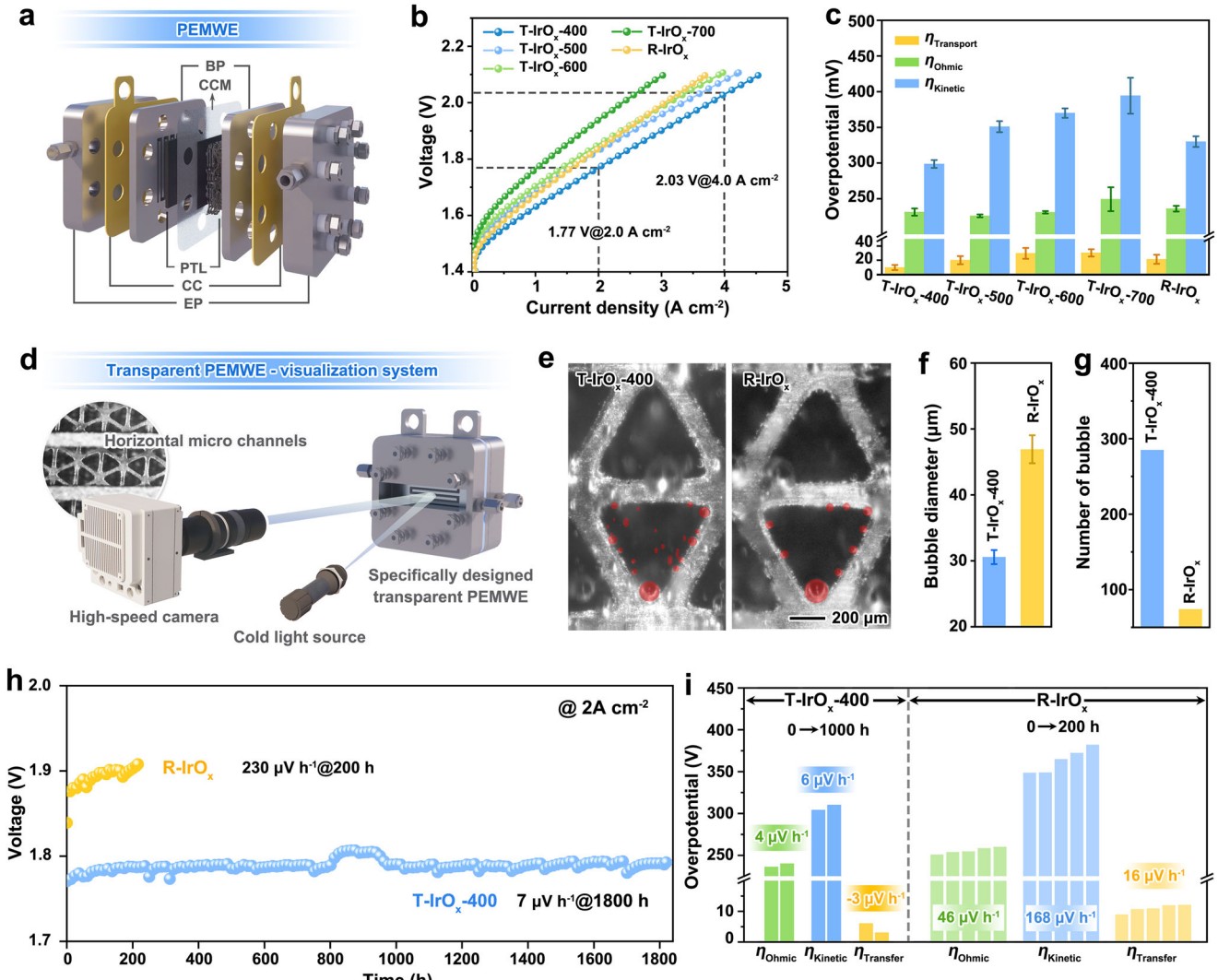

**Fig. 6 | Performance of T-IrOx-based proton exchange membrane water electrolyzers (PEMWE).** **a** Schematic of PEMWE configuration (EP end plate, CC current collector, BP bipolar plate, PTL porous transport layer, CCM catalyst-coated membrane). **b** Polarization curves of PEMWE without iR-corrected employing four T-IrOx samples and R-IrOx as the anode catalysts. **c** Voltage losses without iR-corrected for four T-IrOx-based and R-IrOx-based PEMWE at 2.0 A cm⁻², including $\eta_{Transport}$, $\eta_{Ohmic}$, and $\eta_{Kinetic}$. The error bars represent the standard deviation from three measurements. **d** Schematic of high-speed microscale visualization system with a reaction-visible PEMWE. **e** High-speed movie snapshots of oxygen evolution reaction in T-IrOx-400- and R-IrOx-based PEMWE at 1.0 A cm⁻². Pseudo-coloring was employed to better visualize the bubbles. **f** Diameter and **g** number of produced O₂ bubbles from T-IrOx-400- and R-IrOx-based PEMWE at 1.0 A cm⁻². Error bars indicate the standard error. **h** Chronopotentiometric curve without iR-corrected of T-IrOx-400-based PEMWE at 2.0 A cm⁻². **i** Comparison of overpotentials in T-IrOx-400 and R-IrOx-based PEMWE under different catalytic durations. Source data are provided as a Source data file.

in Supplementary Table 11[41,53–62]. Notably, this non-supported IrOx-based PEMWE is among the few reported systems that meet the 2025 targets of the U.S. Department of Energy in terms of catalytic activity and noble metal loading (≥1.9 V@3.0 A cm⁻², PGM loading ≤0.5 mg cm⁻²)[63].

The performance difference of PEMWE prompts a detailed analysis of the voltage losses. The voltage losses for the four T-IrOx- and R-IrOx-based CCMs are measured by combining electrochemical impedance spectroscopy and steady-state polarization curve measurements (Fig. 6b and Supplementary Fig. 49). The total voltage loss is deconvoluted into transport, ohmic, and kinetic contributions (Supplementary Fig. 50). Figure 6c lists the corresponding overpotentials for each electrolyzer at 2.0 A cm⁻² current density. Both T-IrOx- and R-IrOx-based PEMWEs show comparable $\eta_{Ohmic}$, which arise from the resistance within each component (i.e., membranes, catalyst layers, porous transport layers, and bipolar plates) and the contact resistance

between them. This similarity can be explained by the fact that the voltage loss analysis is performed using the same single-cell PEMWE system under identical conditions. $\eta_{Kinetic}$ originates from the OER kinetics of the catalyst, with lower $\eta_{Kinetic}$ values indicating higher OER activity. It is observed that $\eta_{Kinetic}$ for T-IrOx-based PEMWEs increases with the catalyst synthesis temperature, and the T-IrOx-400-based PEMWE exhibits a lower $\eta_{Kinetic}$ compared to the R-IrOx-based PEMWE. Furthermore, the T-IrOx-400-based PEMWE also exhibits a lower $\eta_{Transport}$ (governed by gas/liquid transport loss) compared to the R-IrOx-based PEMWE. This implies the better gas-liquid transport channels within the T-IrOx-400 catalyst layer, thanks to the high porosity of its nanorod-interwoven architecture.

To further investigate mass transport behavior, we construct a transparent reaction-visible PEMWE, equipped with a thin GDL featuring through-holes, and use a high-speed microscale visualization system to monitor the PEMWE operation in real time (Fig. 6d and

Supplementary Fig. 51). The Supplementary Movie 1 and Supplementary Movie 2 show the oxygen generation and flow paths of the T-IrO$_x$-400- and R-IrO$_x$-based CCMs at the same current density (1.0 A cm$^{-2}$). The T-IrO$_x$-400-based CCM produces oxygen bubbles that are smaller in diameter and more numerous, which quickly leave the catalytic region. As shown in the representative images from the high-speed movie (Fig. 6e), the T-IrO$_x$-400-based PEMWE generates 286 bubbles within 33 ms, with an average radius of 31 µm (Fig. 6f, g). In contrast, the oxygen produced by the R-IrO$_x$-based CCM preferentially adheres to the triangular openings of the PTL, where the bubbles tend to coalesce into larger ones before detaching. This results in fewer bubbles (74) with a larger average radius of 47 µm for the R-IrO$_x$-based CCM. These results further highlight the notable mass transport properties of the T-IrO$_x$-400-based CCM.

The durability of the T-IrO$_x$-400-based PEMWE was assessed through chronopotentiometric measurements at 2.0 A cm$^{-2}$ current density. As shown in Fig. 6h, the T-IrO$_x$-400-based PEMWE exhibited notable stability at an iridium loading of ~0.28 mg cm$^{-2}$. The T-IrO$_x$-400-based single-cell PEMWE exhibited minimal degradation over 1800 h, with an average degradation rate of about 7 µV h$^{-1}$. In contrast, the R-IrO$_x$-based PEMWE with the same low Ir loading exhibited a large degradation rate of 230 µV h$^{-1}$, 32 times higher than that of the T-IrO$_x$-400-based PEMWE. Remarkably, the T-IrO$_x$-400-based PEMWE exhibited notable durability compared to other PEMWEs with representative reported anode catalysts (Supplementary Table 11). On the one hand, compared to T-IrO$_x$-400-based PEMWE, Ir black or IrO$_x$-based CCMs typically exhibited higher degradation rates (≥20 µV h$^{-1}$) even at higher Ir loadings (≥0.5 mg cm$^{-2}$). On the other hand, PEMWEs based on supported iridium catalysts also displayed higher degradation rates (≥88 µV h$^{-1}$) than the T-IrO$_x$-400-based PEMWE, when comparable low Ir loadings are used. These results highlight the strong adaptability of the T-IrO$_x$-400-based CCM for practical applications that require reduced iridium usage, particularly under ampere-level current densities.

To elucidate the degradation mechanisms of the CCMs, a voltage loss analysis on the PEMWEs is conducted after prolonged operation. As shown in Fig. 6i and Supplementary Figs. 52, 53, after 1000 h of operation, T-IrO$_x$-400-based PEMWE exhibits minimal degradation in both $\eta_{Kinetic}$ (6 µV h$^{-1}$) and $\eta_{Ohmic}$ (4 µV h$^{-1}$), as well as a slight improvement in $\eta_{Transport}$ (−3 µV h$^{-1}$), further confirming its high stability. In contrast, during the 200-h operation of the R-IrO$_x$-based PEMWE, all three overpotentials exhibit continuous degradation, with the degradation rates of 180 µV h$^{-1}$ for $\eta_{Kinetic}$, 46 µV h$^{-1}$ for $\eta_{Ohmic}$, and 4 µV h$^{-1}$ for $\eta_{Transport}$. The SEM image of the R-IrO$_x$-based CCM after 200 h of operation shows significant agglomeration of R-IrO$_x$ nanoparticles (Supplementary Fig. 54). As depicted in Fig. 1a, this catalyst migration could lead to the substantial increase in $\eta_{Kinetic}$, $\eta_{Ohmic}$ and $\eta_{Transport}$. To be specific, agglomeration of R-IrO$_x$ nanoparticles reduces the available active surface area on the catalyst layer, renders the formation of electrical dead zones (i.e., islanding effect), and obstructs the gas/liquid transport channels, which contribute to significant increase in $\eta_{Kinetic}$, $\eta_{Ohmic}$, and $\eta_{Transport}$, respectively. In contrast, the 1D morphology and interwoven stacking of the T-IrO$_x$-400 nanorods in the corresponding CCM play a pivotal role in maintaining the structural integrity of the anode catalyst layer, as evidenced by SEM images of post-operation R-IrO$_x$-based CCM (Supplementary Fig. 55). This architecture supports long-term operation through three primary mechanisms (Fig. 1b): (i) preventing catalyst migration, thereby avoiding the shielding of active sites; (ii) maintaining an efficient conductive network to prevent the formation of electrical dead zones; (iii) ensuring the abundant and efficient mass transport pathways.

## Discussion

In summary, our multiscale investigation establishes that the site-specific catalytic activity of tunnel-structured IrO$_x$ (T-IrO$_x$)—with

tunnel mouths exhibiting 25-fold higher oxygen evolution reactivity than tunnel walls—dictates both fundamental efficiency limits and practical optimization pathways. By exploiting this spatial activity heterogeneity through morphology engineering, we develop short T-IrO$_x$ nanorods that increase access to catalytically favorable tunnel-mouth sites while preserving structural integrity. When deployed in PEMWE anodes at low iridium loading (0.28 mg$_{Ir}$ cm$^{-2}$), these 1D nanocatalysts uniquely reconcile three critical yet often conflicting requirements: high intrinsic activity through atomic-site optimization, durable conductive networks via close contact between nanorods, and rapid mass transport enabled by abundant porosity. The resultant electrolyzers achieve notable current densities (>2.0 A cm$^{-2}$@1.8 V) with 1800 h stability, outperforming state-of-the-art R-IrO$_x$ systems. In addition to spatial engineering of active sites, atomic-level compositional tuning, such as Ir-Ru mixing reported by Park et al., represents a complementary strategy for further enhancing catalytic performance[64]. A dual-modulation approach that integrates spatial control with compositional optimization may provide a robust blueprint for developing next-generation tunnel-structured electrocatalysts.

## Methods

### Chemicals and reagents

Potassium hexachloroiridate(IV) (K$_2$IrCl$_6$, 99.99%) was purchased from Hefei Conservation of Momentum Green Energy Co., Ltd. Potassium carbonate (K$_2$CO$_3$, 99%) was purchased from Tianjin Guangfu Technology Development Co., Ltd. Iridium dioxide (IrO$_2$, 99%) and heavy-oxygen water (H$_2{}^{18}$O) with an isotopic purity of 98 atom% $^{18}$O were purchased from Energy Chemical Co., Ltd. Perchloric acid (HClO$_4$, 70.0–72.0%) was purchased from Shanghai Wokai Biotechnology Co., Ltd. Isopropyl alcohol ((CH$_3$)$_2$CHOH) and absolute ethanol (C$_2$H$_5$OH) were purchased from Sinopharm Chemical Reagent Co., Ltd. Nafion® perfluorinated resin solution was purchased from Sigma-Aldrich. Ultrapure water (>18 MΩ cm resistivity) was supplied with a PALL PURELAB Plus system.

### Synthesis of T-IrO$_x$

To prepare the precursor, a conventional solid-state method was employed. Specifically, 30 mg of K$_2$IrCl$_6$ and 35 mg of K$_2$CO$_3$ were thoroughly mixed in an agate mortar and ground for 30 min to ensure homogeneity. The resulting mixture was then transferred to a muffle furnace, heated to 400 °C at a rate of 5 °C min$^{-1}$, and calcined for 2 h. After natural cooling to room temperature, the reaction yielded K$_{0.25}$IrO$_2$ along with KCl as a byproduct. The obtained precursor was subsequently immersed in 1 M HClO$_4$ for 24 h to facilitate proton exchange and remove any residual KCl. The precipitate was then washed three times with deionized water and ethanol, followed by drying at 80 °C. The final black powder was designated as T-IrO$_x$-400.

For comparison, additional samples were synthesized using the same procedure but with calcination temperatures of 500 °C, 600 °C, 700 °C to obtain T-IrO$_x$-500, T-IrO$_x$-600, T-IrO$_x$-700, respectively.

### Material characterizations

X-ray diffraction (XRD) patterns were collected using a Bruker D8 Advance diffractometer. Transmission electron microscopy (TEM) and high-resolution TEM (HRTEM) images were acquired with a Talos F200s G2 TEM, equipped with a field emission gun operating at 200 kV. High-angle annular dark field scanning TEM (HAADF-TEM) images were obtained using a Thermo Spectra 200 microscope with a double tilt holder, also operating at 200 kV. Scanning electron microscope (SEM) images were obtained with field emission scanning microscopy (JEOL JSM-7800F). Inductively coupled plasma atomic emission spectroscopy (ICP-OES) was conducted using a PerkinElmer Optima 3300DV ICP spectrometer. Raman spectra were recorded with a Renishaw Raman System 1000 spectrometer, utilizing a 20 mW air-cooled argon-ion laser (532 nm) as the excitation source. All the

electrochemical measurements in liquid half-cell were carried out on CHI 760E (ChenHua Instrument, Inc., Shanghai). Differential electrochemical mass spectrometry (DEMS) measurements were carried out on an in situ electrochemical mass spectrometer QMG 250 manufactured by Linglu Instruments (Shanghai) Co. Ltd. All the electrochemical measurements in proton exchange membrane water electrolyzer (PEMWE) were carried out on cell fixture purchased from Hefei Conservation of Momentum Green Energy Co., Ltd. The polarization curves of PEMWEs were acquired using a Gamry instrument with a 30A booster under linear sweep scanning conditions. The stability analysis for PEMWE was conducted by employing NEWARE Battery Test System (CT-4008-5V100A, Shenzhen, China) at a water flow rate of 60 mL min⁻¹. The noble metal mass loading was measured using the X-ray fluorescence spectrometer (XRF) manufactured by Shanghai Qinzhi Industrial Co., Ltd.

### Electrochemical measurements in liquid half-cell

Electrochemical measurements in a 30 mL liquid half-cell were performed using a standard three-electrode setup in 0.1 M $HClO_4$, saturated with $O_2$. Platinum wire and saturated calomel electrode (SCE) were used as counter electrode and reference electrode, respectively. Glassy carbon electrode (GCE), platinum wire and saturated calomel electrode were purchased from ChenHua Instrument, Inc., Shanghai. 0.1 M $HClO_4$ electrolyte (measured pH $0.977 \pm 0.006$, determined from three parallel measurements, and reported as pH 1 throughout this work) was prepared by diluting perchloric acid (70.0–72.0 wt%) with deionized water to the desired concentration. All electrolytes were freshly prepared or used within 3 days to ensure chemical stability. The prepared solutions were stored at room temperature in the dark. Unless otherwise specified, all measurements were carried out at room temperature (298 K).

The preparation of the working electrode was as follows: 4 mg of the catalyst powder was dispersed in 400 μL of 0.15% Nafion® solution in isopropanol and ultrasonicated to form a uniform mixture. Subsequently, 2 μL of the catalyst ink was drop-casted onto the GCE (test area: 0.071 cm²), yielding a mass loading of 0.281 mg cm⁻². The electrode was then air-dried at room temperature for further use. Linear sweep voltammetry (LSV) was performed between 1.23 and 1.55 V vs. RHE at the scanning rate of 1 mV s⁻¹ with 85% iR-correction. iR compensation for LSV measurements was applied via positive feedback, following the protocol established by Jaramillo et al.[65,66].

The SCE was calibrated against a reversible hydrogen electrode (RHE) following a modified procedure based on a previously reported method[67]. Briefly, two Pt electrodes were first cleaned by cyclic voltammetry (CV) in 1 M $H_2SO_4$ between −2.0 and 2.0 V for 2 h. Afterward, they were assembled as the working and counter electrodes in a three-electrode configuration. The electrolyte (0.1 M $HClO_4$) was saturated with high-purity $H_2$ gas for at least 30 min prior to calibration and continuously bubbled during the measurement. The equilibrium potential was determined using the "open-circuit potential" function of the electrochemical workstation, which records the potential at which the net current between hydrogen evolution and oxidation is zero. In our setup, the equilibrium potential of the RHE was found to be −0.306 V versus SCE in 0.1 M $HClO_4$. Therefore, all potentials measured against the SCE were converted to the RHE scale using the Eq. (1):

$$E_{RHE} = E_{SCE} + 0.306 \tag{1}$$

Geometric current density ($j_{geo}$) of catalyst was normalized by the geometric area of GCE according to Eq. (2):

$$j_{geo} = \frac{i \times 1000}{S} \tag{2}$$

where $i$ (A) is the current with 85% iR-correction, and $S$ is the geometric area of GCE (0.071 cm²).

The intrinsic activity ($j_Q$) of the catalyst was normalized with the pseudocapacitive charge ($Q$) according to Eq. (3):

$$j_Q = \frac{i}{Q} \tag{3}$$

where $Q$ (C) is the pseudocapacitive charge, calculated by integrating the cathode or anode currents from CV curves. CV measurement was conducted in the potential range of 0–1.4 V at a scanning rate of 50 mV s⁻¹[68–70].

To evaluate the electrochemically active surface area (ECSA), CV measurements were performed at scan rates of 20, 40, 60, 80, and 100 mV s⁻¹ within the potential window of 0.81–0.91 V vs. RHE. The geometric double-layer capacitance ($C_{dl}$) was determined by plotting the average current density difference $\Delta J = (J_{anodic} - J_{cathodic})/2$ at 0.86 V vs. RHE as a function of the scan rate, with the slope of the resulting linear fit corresponding to $C_{dl}$. The ECSA was estimated using the Eq. (4):

$$ECSA = \frac{C_{dl}}{C_s} \times S \tag{4}$$

where $C_s$ is the specific capacitance per unit area, estimated to be 0.06 mF cm⁻² in this study. $S$ is the geometric area of GCE (0.071 cm²). The current density values ($j_{ECSA}$) were further normalized by the calculated ECSA according to Eq. (5):

$$j_{ECSA} = \frac{i \times 1000}{ECSA} \tag{5}$$

In situ DEMS measurements were carried out by coupling a three-electrode cell with a mass spectrometer (Linglu Instruments (Shanghai) Co. Ltd) in 0.1 M $HClO_4$. The membrane used for DEMS measurements was a hydrophobic gas-permeable PTFE membrane (pore size ≤20 nm, porosity ≥50%), purchased from Linglu Instruments (Shanghai) Co., Ltd. The membrane was not undergone any pretreatment. Firstly, we labeled the catalysts with $^{18}O$ isotope by chronoamperometric measurement at 1.65 V vs. RHE for 10 min in $^{18}O$-labeled 0.1 M $HClO_4$. Next, the labeled electrode was thoroughly washed with $H_2^{16}O$ to remove the physically attached $H_2^{18}O$ on the catalyst surface. Subsequently, CV measurement was conducted in the potential range of 0.6–1.2 V at a scanning rate of 50 mV s⁻¹ to further eliminate the $^{18}O$ substance adsorbed on the catalyst. Lastly, LSV test was performed three times in 0.1 M $HClO_4$ containing $H_2^{16}O$ in the potential range of 0.8–1.65 V vs. RHE. Meanwhile, $O_2$ products ($^{32}O_2$, $^{34}O_2$, and $^{36}O_2$) were monitored by mass spectrometer, and were quantified based on the integrated area of signal intensity of gas products. The ion current signals shown were not subjected to deconvolution.

To measure the Faradaic efficiency, the actual amount of $O_2$ produced during the OER process at 30 mA cm_geo⁻² current density was measured by drainage method. Then, according to Faraday's law, the theoretical amount of $O_2$ was computed, assuming that 100% of the current was used for $O_2$ production. Finally, the Faradaic efficiency of the catalyst was determined by calculating the ratio of the measured $O_2$ to the theoretical $O_2$.

### Electrochemical measurements in PEMWE

To prepare for catalyst-coated membrane (CCM) fabrication, the Nafion® 115 membrane (N115, Chemours) was treated sequentially with 3 wt% $H_2O_2$, 0.5 M $H_2SO_4$, and deionized water, each step lasting 1 h at 80 °C. The processed N115 was subsequently stored in deionized water following the treatment process. T-IrOₓ was used as the anode catalyst

and commercial Pt/C (40 wt%) was used as the cathode catalyst. To prepare the catalyst ink, catalysts were ultrasonically dispersed in a mixed solution of isopropyl alcohol and deionized water (1:1, w/w), and 5 wt% Nafion® solution was added. The mass fractions of ionomer relative to the total weight of anode and cathode were 17 wt% and 35 wt%, respectively. Cathode and anode catalyst inks were sprayed onto both sides of N115, respectively, using an ultrasonic spraying instrument. Subsequently, the catalyst-supported N115 was hot-pressed for 3 min at 130 °C under a pressure of 10 MPa, yielding the CCM after cooling. The mass loading of Ir and Pt was controlled at 0.28 mg cm$^{-2}$ and 0.21 mg cm$^{-2}$, respectively, as verified by XRF.

To construct the PEMWE, a single-cell fixture was used, featuring a CCM with an active area of 5 cm$^2$. The anode and cathode were equipped with a piece of 0.25 mm thick titanium felt coated with a 2 μm layer of platinum and a piece of 0.2 mm thick carbon paper as porous transfer layer (PTL), respectively. Before testing, the PEMWE was continuously flushed with deionized water at 80 °C for at least 5 h using a circulating peristaltic pump. To activate the CCM, it was tested at constant current densities of 0.1 A cm$^{-2}$ and 0.5 A cm$^{-2}$, with each condition maintained for 1 h. The polarization curve was obtained by scanning at a rate of 20 mV s$^{-1}$ over a potential range of 1.4–2.1 V. Stability was assessed using chronopotentiometry, applying current density of 2.0 A cm$^{-2}$.

Cell voltage ($E_{cell}$) consists of reversible cell voltage ($E_0$) and three types of overpotentials: ohmic overpotential ($\eta_{Ohmic}$), kinetic overpotential ($\eta_{Kinetic}$) and mass transport overpotential ($\eta_{Transport}$). $E_{cell}$ can be expressed by Eq. (6)[71,72]:

$$E_{cell} = E_0 + \eta_{Ohmic} + \eta_{Kinetic} + \eta_{Transport} \qquad (6)$$

Ohmic resistance arises from the resistance in electrode, electrolyzer, and proton-electron transport through membrane electrode assembly (MEA). The value of $\eta_{Ohmic}$ can be calculated using Ohm's law, expressed by Eq. (7):

$$\eta_{Ohmic} = j \times HFR \qquad (7)$$

where $j$ represents current density (A cm$^{-2}$), and HFR is high-frequency resistance, which is from the electrochemical impedance spectroscopy (EIS, Supplementary Fig. 42).

The $\eta_{Kinetic}$ arises from direct electron transfer between redox couples at the electrode-electrolyte interface during OER and HER, and its magnitude depends on the intrinsic activity of catalyst. $\eta_{Kinetic}$ can be calculated using Eq. (8):

$$\eta_{Kinetic} = b \times \log\left(\frac{j}{j_0}\right) \qquad (8)$$

where $b$ is Tafel slope, a parameter that is obtained by fitting polarization curve after correcting for ohmic losses, particularly at low current densities. $j_0$ is the exchange current density (A cm$^{-2}$), determined by extrapolating the Tafel slope.

Mass transport losses arise from diffusion limitations of reactant water and product gases, with retained bubbles, which may result in blocking the electrochemically active sites. $\eta_{Transport}$ can be obtained from Eq. (9):

$$\eta_{Transport} = E_{Ohmic} - \text{corrected} - \eta_{Kinetic} \qquad (9)$$

where $E_{ohmic-corrected}$ is obtained by correcting $E_{cell}$ for ohmic resistance.

The high-speed and microscale visualization system (HMVS) system includes a high-speed camera (Phantom VEO711) and a long-distance microscope (Infinity Model K2 DistaMaxTM) (Supplementary Fig. 44). The high-speed camera can achieve a frame rate of up to

7500 fps at maximum resolution and 600,000 fps at lower resolutions. The long-distance microscope, equipped with a main zoom lens body, various objective and eyepiece lenses, provides a working distance of >50 mm even at high resolution. This feature distinguishes it from traditional microscopes and provides sufficient space for observing mass transport in PEMWE. The relative distance between the reaction-visible PEMWE and the HMVS can be finely adjusted. All components are arranged on an anti-vibration optical table to ensure stability and accuracy. Additionally, an adjustable-intensity cold light source, delivered through gooseneck probes, ensures high-quality movies and images.

## Computational methods

The Vienna ab initio simulation package (VASP 5.4.4) was used to perform all density functional theory (DFT) computations[73,74]. The Perdew-Burke-Ernzerhof (PBE) functional within the generalized gradient approximation (GGA) was used to describe the exchange-correlation interaction[75,76]. The electronic convergence criterion was 10$^{-4}$ eV, while the force criterion for geometry relaxation was 0.02 eV Å$^{-1}$. The cutoff energy of the plane wave basis set was set to 500 eV. Gamma-centered k-point grids were used for all calculations, and the symmetry was turned off. A k-point separation length of 0.04 2π Å$^{-1}$ was used for all structure optimizations, and 0.03 2π Å$^{-1}$ was used for DOS calculations[77]. Slab models were constructed with the help of AFLOW and pymatgen[78–80]. For all slabs, the bottom half along the vertical $z$-direction was constrained, while the top half and the adsorbed species were relaxed. A 15 Å vacuum layer was added in the $z$-direction, and a dipole correction was applied. Grimme's DFT-D3 method was used for the van der Waals (vdW) corrections[81,82]. The VASPKIT code was used to generate all **k**-points and analyze the VASP computational data, and LOBSTER was used for the crystal orbital Hamiltonian populations (COHP) calculations[83–86]. For equivalent crystal planes with the same surface, only one of them was considered. For example, in the case of (100) surface and (010) surface in 2 × 2 tunnel, we only calculated the (100) surface. All computational structure files are provided in Supplementary Data 1.

## Theoretical evaluation of ion leaching

The exchange between metal cation and proton in tunnel-structured iridates and free energy change of this process ($\Delta G_{leaching}$) can be written as Eqs. (10 and 11), similarly to the previous work by Jaramillo et al.[36]

$$A_xIrO_{2(s)} + xH^+_{(aq)} \rightarrow H_xIrO_{2(s)} + xA^+_{(aq)} \qquad (10)$$

$$\Delta G_{leaching} = \mu_{H_xIrO_2} + x\mu_{A^+} - \mu_{A_xIrO_2} - x\mu_{H^+} + xk_BT \ln\frac{[A^+]}{[H^+]} \qquad (11)$$

where $\mu_{H_xIrO_2}$, $\mu_{A^+}$, $\mu_{A_xIrO_2}$, and $\mu_{H^+}$ are chemical potential of H$_x$IrO$_2$, A$^+$, A$_x$IrO$_2$, and H$^+$ respectively. $k_B$ is Boltzmann constant (1.38 × 10$^{-23}$ J K$^{-1}$).

We approximated the chemical potential of solids as total energy from DFT and use the electrochemical series to eliminate A$^+$ as shown in Eqs. (12–14):

$$\mu_{A^+} = \mu_A - \mu_{e^-} + eE_{A^+/A} \qquad (12)$$

$$\mu_{H^+} = 0.5\mu_{H2} - \mu_{e^-} \qquad (13)$$

$$\mu_{H_xIrO_2} - \mu_{A_xIrO_2} \approx E^{DFT}_{H_xIrO_2} - E^{DFT}_{A_xIrO_2} \qquad (14)$$

where $\mu_A$ is chemical potential of neutral metal A, $e$ is elementary charge, and $E_{A^+/A}$ is standard electrode potential of the A/A$^+$ redox couple.

Therefore, $\Delta G_{leaching}$ can be written as shown in Eq. (15):

$$\Delta G_{leaching} = E_{H_xIrO_2}^{DFT} + xE_A^{DFT} - E_{A_xIrO_2}^{DFT} - \frac{1}{2}x\mu_{H_2} + eE_{A/A^+} + xk_BT\ln\frac{[A^+]}{[H^+]}$$

$$(15)$$

## Theoretical OER activity

The theoretical OER activity was calculated based on the adsorbate evolution mechanism (AEM) and lattice oxygen mechanism (LOM), which are shown in Eqs. (16–24).

AEM:

$$H_2O + * \rightarrow OH* + H^+ + e^- \qquad (16)$$

$$OH* \rightarrow O* + H^+ + e^- \qquad (17)$$

$$H_2O + O* \rightarrow OOH* + H^+ + e^- \qquad (18)$$

$$OOH* \rightarrow O_2 + * + H^+ + e^- \qquad (19)$$

LOM:

$$H_2O + * \rightarrow OH* + H^+ + e^- \qquad (20)$$

$$OH* \rightarrow O* + H^+ + e^- \qquad (21)$$

$$O^* \rightarrow O_V^* + O_2 \qquad (22)$$

$$O_V^* + H_2O \rightarrow H^* + H^+ + e^- \qquad (23)$$

$$H^* \rightarrow * + H^+ + e^- \qquad (24)$$

where * represents the active site of the catalyst and $O_V^*$ represents the catalyst surface containing an oxygen vacancy.

Using a CHE model, the reaction free energies for each step of AEM can be calculated as shown Eqs. (25–28)[87].

$$\Delta G_1 = \Delta G_{OH^\cdot} + 0.5\Delta G_{H_2} - \Delta G_{H_2O} - \Delta G^* - eU \qquad (25)$$

$$\Delta G_2 = \Delta G_{O^\cdot} + 0.5\Delta G_{H_2} - \Delta G_{OH^\cdot} - eU \qquad (26)$$

$$\Delta G_3 = \Delta G_{OOH^\cdot} + 0.5\Delta G_{H_2} - \Delta G_{H_2O} - \Delta G_{O^\cdot} - eU \qquad (27)$$

$$\Delta G_4 = \Delta G_{O_2} + 0.5\Delta G_{H_2} + \Delta G^* - \Delta G_{OOH^\cdot} - eU \qquad (28)$$

The free energy change was calculated using Eq. (29):

$$\Delta G = \Delta E_{DFT} + T\Delta S + \Delta ZPE \qquad (29)$$

where $\Delta E_{DFT}$ (eV) is the energy change calculated using VASP, $\Delta S$ (eV K$^{-1}$), and $\Delta ZPE$ (eV) are the entropy change and zero-point energy change, respectively. Here, $T$ is the temperature set to 298 K. The calculated overpotential is defined as Eq. (30):

$$\eta_{OER} = \max(\Delta G_{1-4})/e - 1.23 \qquad (30)$$

Although the CHE model does not explicitly capture the influence of interfacial solvation or potential-dependent electronic effects, it provides reliable qualitative trends when comparing analogous materials. For a more rigorous treatment of these interfacial phenomena, beyond-CHE approaches such as constant-potential simulations and explicit solvation models have been proposed[88,89], which represent promising future directions complementary to the CHE-based approach used in this work.

The energy for $O_2$ was derived from the reaction $2H_2O \rightarrow 2H_2 + O_2$, in which 4.92 eV of energy is required to form one molecule of oxygen (Supplementary Table 12).

## Surface energy

Surface energy was calculated as follow Eq. (31):

$$\gamma = \frac{1}{2A}(E_s^{unrelax} - nE_b) + \frac{1}{A}(E_s^{relax} - E_s^{unrelax}) \qquad (31)$$

where A was the surface area (m$^2$), $E_s^{unrelax}$ and $E_s^{relax}$ (eV) were the total energy of unrelaxed and relaxed slab, respectively, which containing $n$ formula units. $E_b$ (eV) was the energy per formula unit of the bulk.

## Data availability

The source data generated in this study are provided in the Supplementary Information and Source Data file. No third-party or sensitive data are involved, and the dataset is not subject to any access restrictions. Source data are provided with this paper.

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

## Acknowledgements

M.Z., W.A., Q.L., and Y.J. contributed equally to this work. X.X.Z. acknowledges the financial support from the National Natural Science Foundation of China (NSFC) (no. 22179046). X.L. acknowledges the financial support from the National Natural Science Foundation of China (NSFC) (no. 22205072). Y.Z. acknowledges the Jilin Province Science and Technology Development Plan (no. 20220203086SF). H.C. acknowledges financial support from the Fundamental Research Funds for the Central Universities. M.Z. acknowledges the financial support from the Graduate Innovation Fund of Jilin University.

## Author contributions

The manuscript was written through the contributions of all authors. All authors discussed and reviewed the final manuscript. M.Z., W.A., Q.L., and Y.J. contributed equally to this work. X.X.Z. and X.L. directed this research. M.Z. and Y.J. performed the theoretical calculations. W.A. and Q.L. conducted most of the experiments. X.Z., Y.Z., and H.C. contributed to data analysis. X.X.Z., X.L., and M.Z. wrote the paper. X.X.Z. and X.L. supervised the project.

## Competing interests

The authors declare no competing interests.
