## [Transparent Peer Review file · Nature Communications]

Tunnel-Structured IrOx Unlocks Catalytic Efficiency in Proton Exchange Membrane Water Electrolyzers

Corresponding Author: Professor Xiaoxin Zou

Version 0:

Reviewer comments:

Reviewer #1

(Remarks to the Author)

Review of "Unlocking catalytic efficiency in PEMWE electrolyzers through spatial control of active sites in tunnel-structured IrOx nanocatalysts" for Nature Communications

Recommendation: Reject

Comments: The authors have synthesized 2x2 tunnel-structured iridium oxide (T-IrOx) nanorods through a low-temperature solid-state reaction followed by acid treatment. Experimental results revealed that T-IrOx nanorods with the highest exposure of tunnel mouths exhibit superior activity toward the oxygen evolution reaction. Theoretical studies further demonstrated the nonuniform activity distribution within T-IrOx, resulting in varying intrinsic activities within the same material system. These findings were reflected in the proton exchange membrane water electrolyzer (PEMWE) performances, achieving current densities exceeding 2.0 A cm^{-2} at 1.8 V with 1600-hour stability, surpassing state-of-the-art rutile IrOx (R-IrOx) systems. However, a more comprehensive discussion on the correlation between tunnel structure characteristics and catalytic properties is required, as the current manuscript lacks sufficient supporting data. Prior to publication in Nature Communications, it is necessary to address the following comments.

1. Figure 3 presents various calculation results demonstrating the difference in OER properties between the tunnel mouths and tunnel walls. However, the wall structure of T-IrOx consists of an inner wall and an outer wall, which have completely different environments. These two structures are expected to have a significant difference in their catalytic effects. The authors need to establish an additional computational model to explain these differences.
2. The tunnel-structured iridium oxide (T-IrOx) nanorods synthesized in this study exhibit a tunnel mouth structure with a 2x2 size and a length of 93 nm. By modifying the experimental conditions, it seems possible to control the size of the tunnel mouth or synthesize shorter rods under lower temperature conditions or different reaction times. The authors should propose additional experiments or include further discussion on this possibility.
3. In the experimental section, the acid treatment in 1M HClO₄ was essential for leaching out K⁺ ions. During this process, it seems that the IrOx tunnel could undergo over-oxidation or dissolution, potentially leading to Ir loss in the catalyst. The author needs to explain these issues.
4. In the OER mechanism, the oxidation state of Ir is a critical factor affecting both OER kinetics and stability. The authors should conduct additional XPS (X-ray photoemission spectroscopy) or XAS (X-ray absorption spectroscopy) analysis to examine the Ir oxidation states in the catalyst before and after electrochemical tests to support the OER performances.
5. The T-IrOx-400 exhibited the highest OER activity compared to other T-IrOx nanorod catalysts in Figure 5. The authors also calculated the intrinsic activity (j_Q) by normalizing the current with respect to the pseudocapacitive charge (Q_s). However, T-IrOx-400, with the smallest aspect ratio, is expected to have the largest catalytic active area. Therefore, additional normalized current densities based on the electrochemically active surface area (ECSA) are required to accurately compare the intrinsic OER activities.
6. According to various experimental and theoretical data presented in this study, the OER activities of the T-IrOx catalyst appear to be strongly dependent on the tunnel mouth structure. However, these results fail to fully explain the advantages of the tunnel structure in T-IrOx catalyst. Without addressing the synergistic effect between the tunnel mouths and tunnel walls, or the catalytic reactions occurring inside the tunnel, the tunnel structure does not seem to offer significant benefits compared

to nanorod or nanoribbon structures. A more detailed and clear explanation of how the tunnel structure influences catalytic performance is required.

Reviewer #2

(Remarks to the Author)

The manuscript reports the development of tunnel-structured IrO₂ catalysts for the oxygen evolution reaction in acidic medium.

I have been asked to assess the computational part, so I will not comment on the experimental achievements.

The first surprise comes in the abstract, where the authors state that the "tunnel mouths" are most active, but that nanorods are most active. One would have expected that plaquettes, rather than rods, should be optimal if the cross-section of the tunnel is most active compared to the outer-walls.

Then, the introduction discusses the limitations of IrO₂ as stemming from 3 aspects: insufficient active sites per surface area, charge-transfer issues and nanoparticle agglomeration.

The computational part of the study (and the resulting rationalisation for the observed performances) do not really come back to these limitations: the charge-transfer is discussed, but the computations do not yield valuable insight on this aspect. Other than that, the computational study claims higher intrinsic activity of the created "mouth" sites. However, according to the introduction, the intrinsic activity of IrO₂ is already good enough, so the computational study seems to miss the point.

Furthermore, not much is said to explain the supposedly significant difference between the T-IrO_x in the literature and the one presented herein: What is the critical difference that activates the current T-IrO_x compared to the ones studied before? This is critical, as the computational study suggest that R-IrO₂ is pretty inactive. Hence there is some contradiction to be dealt with.

The computational method is rather crude, in the sense that it does not include the explicit consideration of the electrochemical potential, nor explicit solvation. The former is key for discussing "protonation" (vs. hydrogenation; this is important in view of the ion-exchange) and the later (at least microsolvation) would be important to know the energy for water to enter into the tunnels.

In summary, the work seems carefully carried out, but overall coherence within the manuscript and the scientific hypothesis/reasoning requires significant improvements before the manuscript could be publishable anywhere.

Details:

- "fundamentally redefining design principles for iridium oxide catalysts": This is a too strong statement compared to the content of the manuscript.

- All the computed structures should be made available in the supplementary information or as a webarchive (typically on nomad-lab.eu). This is key for reproducibility.

- An OER overpotential of 0.5 V should not be qualified as "excellent".

Reviewer #3

(Remarks to the Author)

The authors postulated through DFT calculations that tunnel mouth sites exhibit higher catalytic activity than tunnel wall sites in tunnel-structured IrO_x (T-IrO_x) materials. To experimentally validate this hypothesis, they synthesized a series of T-IrO_x-XXX nanorod samples by modulating synthesis temperatures to control the exposure ratio of tunnel mouth sites. The resultant T-IrO_x-400 sample applied in PEM electrolyzers achieve activity and outstanding long-term durability at large current density 2 A cm⁻². They proposed that the superior performance of T-IrO_x-400 was attributed to its shortened nanorod morphology, which purportedly maximizes the density of active tunnel mouth sites. However, a critical question arises regarding the potential confounding influence of structural defects. The relatively low synthesis temperature (400°C) may result in reduced crystallinity, as evidenced by the broadened diffraction peaks in the XRD patterns (Fig. 4b). This raises the possibility that the enhanced activity of T-IrO_x-400 could stem from defect-rich surfaces rather than the proposed tunnel mouth site dominance. The authors should provide more evidence to decouple the contributions of crystallographic defects from tunnel mouth site exposure (e.g., operando XPS for oxygen vacancy quantification, or operando EXAFS for local disorder assessment).

Overall, this study presents an innovative approach to probing the relationship between distinct tunnel structures and intrinsic activity. However, major revisions are required to address the aforementioned questions and several important issues.

Other issues:

1. In the electrochemical LSV characterization (Figure 5a), T-IrO_x-400 shows enhanced intrinsic activity compared to other higher temperature-controlled counterparts. However, critical evaluation of operational durability remains incomplete. While the manuscript demonstrates promising stability for T-IrO_x-400 through chronopotentiometric testing, the absence of comparative stability data for other temperature-varied samples weakens the performance superiority of T-IrO_x-400. To enhance the comprehensive evaluation, the stability data for other temperature-controlled variants should be provided.

2. The manuscript defines the "mouth area ratio" in Figure 5b as the ratio of the bipyramidal area to the total surface area of the nanorod. To improve reader comprehension, I recommend including a step-by-step calculation example using a representative nanorod to illustrate this concept more concretely.

3. The XRD patterns presented in Figure 4b and Figure S11 currently only indicate the crystallographic planes corresponding to each diffraction peak. To enhance the structural characterization, I recommend identifying the crystal

system and space group through comparison with standard PDF card. Or, explicit references could be provided to substantiate the phase identification.

4. The Raman spectra shown in Supplementary Fig. 25 exhibit two characteristic peaks at around 550 and 720 cm^{-1} . To enhance the readers' understanding of the tunnel-structured IrOx materials, the authors should provide detailed assignment of these two spectral features.

5. In the PEMWE section, the authors present an SEM image of R-IrOx nanoparticles after prolonged operation in Supplementary Figure 36, giving the observed degradation in R-IrOx morphology. To enable a meaningful comparison of catalyst durability, it would be valuable to include corresponding post-operation SEM of T-IrOx after extended PEMWE operation.

Version 1:

Reviewer comments:

Reviewer #1

(Remarks to the Author)

Review of "Unlocking catalytic efficiency in PEM electrolyzers through spatial control of active sites in tunnel-structured IrOx nanocatalysts" for Nature Communications.

Recommendation: Minor Revision

Comments:

The authors have made significant revisions in response to the initial reviewer feedback. It is clear that several key points raised in the first round have been carefully considered and addressed. The additional experimental data and analyses have notably improved the clarity and robustness of the study. The study's core message, focusing on the spatially localized catalytic activity at the tunnel-mouth regions of IrOx nanorods through morphological control, is now more clearly emphasized. This enhanced presentation is well supported by the newly added operando and structural characterization results. Addressing the following comments will further strengthen the mechanistic insights and overall contribution, enhancing the manuscript's suitability for publication in Nature Communications.

1. Clarification of ECSA-normalized intrinsic activity (jECSA):

One key point raised in the initial review was the importance of normalizing catalytic activity to the electrochemically active surface area (jECSA), particularly in light of the substantial variation in nanorod aspect ratios and morphologies across the T-IrOx series. While the authors continue to rely primarily on pseudocapacitive charge normalization (jQ), the manuscript would benefit from explicitly reporting jECSA values and providing a direct comparison between jQ and jECSA trends. Such a comparison would help determine whether the observed performance enhancements originate from genuine improvements in intrinsic activity or from increased surface accessibility. If experimental limitations prevent the inclusion of jECSA, a discussion of potential discrepancies and their implications for interpreting the structure–activity relationship would still be valuable.

2. Kinetic analysis via Tafel slope comparison:

We appreciate the authors' detailed characterization of intrinsic catalytic activity and the comprehensive presentation of activity data. However, the manuscript currently lacks Tafel slope analysis, which is essential to deepen mechanistic understanding. Given the study's focus on morphological effects, comparing Tafel slopes across the T-IrOx series and R-IrOx samples would provide valuable insight into how tunnel-mouth exposure influences OER kinetics. This analysis could clarify whether tuning the nanorod aspect ratio affects the rate-determining step or overall reaction mechanism. Including Tafel slope data will complement the existing results by revealing changes in reaction kinetics, thereby strengthening the interpretation of morphology-dependent catalytic behavior.

3. Structural stability of nanorod tips and clarification of TEM image assignment:

Regarding structural stability, the revised manuscript more clearly emphasizes the long-term durability of T-IrOx-400, including the newly added SEM image of the catalyst layer after prolonged PEMWE operation (Supplementary Fig. 48), which shows that the interwoven nanorod network remains intact with no observable collapse or detachment. This addition addresses earlier concerns about overall structural degradation. However, previously raised questions about localized deformation at the nanorod tips—identified as key catalytic hotspots—still warrant further clarification. Supplementary Fig. 37 shows visible tip deformation after extended operation, contrasting with the pristine morphology in Supplementary Fig. 15. The authors should clarify whether these changes involve superficial rounding, amorphization, or facet reorganization, and whether such alterations affect the integrity or activity of the exposed active sites. Providing such clarification would help reconcile the apparent structural changes with the sustained catalytic performance. Additionally, the sample shown in Supplementary Fig. 15 should be explicitly identified, as the tip morphology closely resembles that of T-IrOx-700 (Fig. 4f), which could cause confusion given the importance of tip structure in the study's design rationale.

4. Additional literature context for comparison of design strategies:

To provide further context and mechanistic perspective, the authors might consider referencing the recent study published in Nature Communications, titled "Atomic-level Ru-Ir mixing in rutile-type (RuIr)O₂ for efficient and durable oxygen evolution catalysis" (Nat. Commun. 16, 579, 2025). This work also investigates nanorod morphology, active site distribution, and catalyst stability under acidic OER conditions. Drawing a brief comparison to this study, e.g., how the present work emphasizes spatial engineering of tunnel mouths rather than atomic-level compositional tuning, would provide useful perspective and help position the current manuscript within the broader literature landscape.

Reviewer #2

(Remarks to the Author)

The authors have performed a reasonably careful revision, but a few issues remain:

- The recent literature is insufficiently cited, e.g., doi/10.1002/sml.202412237 and, for the modelling part, DOI10.1039/d2ee00158f and 10.1002/wcms.1499, emphasizing the need to go beyond CHE in order to avoid over-simplified conclusions.
- The introduction/motivation of the study is still partially at odds with the final claims/conclusions: T-IrO₂ has “technical” advantages that could address the challenges stated on the top of the second page of the introduction. However, there seems to be no need for higher activity compared to R-IrO₂: Fig 3c reveals that 6-7 active sites on the tunnel walls should be as active as R-IrO₂, so that the poor performance of “standard” (high aspect ratio) remains a bit surprising. Furthermore, it should be clarified that the overpotential of R-IrO₂ is not the main issue and it is just a lucky coincidence that the mouths are a bit more active (Fig 6c), but it is not responsible for the main importance of the current study. Along the same lines, it might be useful to rephrase “with short nanorods achieving optimal balance between active site exposure and electron/mass transport efficiency”: According to my understanding, the authors have no means to claim “optimal balance”: The boundary (the shortest nanorods that are achievable) are the most active, but they have no idea if even shorter nanorods would not perform even better. To summarize this somewhat vague point, I would like to invite the authors to tune down the claims and focus on the main message: shorter rods are more active and still stable. This applies to the main text and the first sentence of the “Discussion”.

After these rather minor modifications, I expect the manuscript to be publishable.

Reviewer #3

(Remarks to the Author)

The authors have addressed all concerns raised during the initial review. The revisions-particularly regarding the possible influence of structural defects on the catalytic performance of T-IrO_x-400 are carefully investigated and discussed through EPR and XAS fitting analysis. The additional experiments/data/explanations satisfactorily resolve the concerns noted earlier to strengthen the manuscript.

The manuscript now meets the scientific rigor and presentation standards required for publication in Nature Communications.

Version 2:

Reviewer comments:

Reviewer #1

(Remarks to the Author)

The work is now suitable for publication.

Reviewer #2

(Remarks to the Author)

The authors have taken appropriate action to resolve the remaining issues.

I have no further comments or requests and recommend publication of this manuscript.

Reviewer's Comments and Our Responses:

Reviewer #1:

Comment 1: The authors have synthesized 2×2 tunnel-structured iridium oxide (T-IrO_x) nanorods through a low-temperature solid-state reaction followed by acid treatment. Experimental results revealed that T-IrO_x nanorods with the highest exposure of tunnel mouths exhibit superior activity toward the oxygen evolution reaction. Theoretical studies further demonstrated the nonuniform activity distribution within T-IrO_x, resulting in varying intrinsic activities within the same material system. These findings were reflected in the proton exchange membrane water electrolyzer (PEMWE) performances, achieving current densities exceeding 2.0 A cm⁻² at 1.8 V with 1600-hour stability, surpassing state-of-the-art rutile IrO_x (R-IrO_x) systems. However, a more comprehensive discussion on the correlation between tunnel structure characteristics and catalytic properties is required, as the current manuscript lacks sufficient supporting data. Prior to publication in Nature Communications, it is necessary to address the following comments.

Response 1: Thanks for the reviewer's evaluation and comments. We note that the question in this comment—specifically, the “correlation between tunnel structure characteristics and catalytic properties”—overlaps with the concerns raised in Comment 7 regarding how the tunnel structure influences catalytic performance. As these are essentially addressing the same issue, we have provided a detailed response under Response 7. Please refer to Response 7 for a full discussion. In the revised manuscript, we have also expanded the relevant discussion to clarify these points.

Comment 2: Figure 3 presents various calculation results demonstrating the difference in OER properties between the tunnel mouths and tunnel walls. However, the wall structure of T-IrO_x consists of an inner wall and an outer wall, which have completely different environments. These two structures are expected to have a significant difference in their catalytic effects. The authors need to establish an additional computational model to explain these differences.

Response 2: We appreciate the reviewer's constructive comments regarding the catalytic differences between the inner and outer walls of the T-IrO_x tunnel structure. This is indeed an important issue that was not fully addressed in our previous calculations. Furthermore, as also raised by Reviewer 2, the question of whether water can enter the tunnel interior is directly relevant to the catalytic behavior of the inner active sites. Therefore, in our revised study, we systematically addressed two related questions:

1. Can water molecules enter the tunnel structure?
2. If so, how active are the inner wall sites under these conditions?

Our calculations show that water cannot enter the 1×2 and 1×2' tunnels due to their small tunnel sizes, which leads to significant steric hindrance. For larger tunnels (2×2, 2×3, and 3×3), water molecules can be accommodated; however, the thermodynamic driving force for water insertion is small (0.02–0.03 eV/atom).

For these larger tunnels where water can enter, we further calculated the OER activity of the inner active sites. The results indicate that for the majority of possible catalytic sites (82%), the key intermediate OOH* cannot be stabilized, meaning the OER cycle cannot be completed. For the remaining sites (18%), the calculated overpotentials exceed 1.5 V, indicating that these inner active sites are theoretically inert.

We have now added a detailed discussion of these results in the revised manuscript (see page 7: “In addition, we evaluated the accessibility of the tunnel interior to water...” and page 8: “The possibility of OER occurring at the inner active sites...”).

Comment 3: The tunnel-structured iridium oxide (T-IrO_x) nanorods synthesized in this study exhibit a tunnel mouth structure with a 2×2 size and a length of 93 nm. By modifying the experimental conditions, it seems possible to control the size of the tunnel mouth or synthesize shorter rods under lower temperature conditions or different reaction times. The authors should propose additional experiments or include further discussion on this possibility.

Response 3: We thank the reviewer for this suggestion. We would like to clarify and apologize for not providing sufficient background on the tunnel mouth sizes in the original manuscript. 2×2 tunnel is the only tunnel-structured iridium oxide that has been experimentally synthesized in pure phase. While other tunnel mouth sizes are theoretically possible, their experimental synthesis is extremely challenging and, to our knowledge, has not been achieved. Therefore, we included them only as theoretical models to investigate the effect of tunnel mouth size on catalytic performance.

Besides, we have systematically studied the effects of synthesis temperature and reaction time. A minimum temperature of 400 °C is required to obtain tunnel-structured IrO_x, because lower temperatures result in amorphous IrO_x. When fixing the temperature at 400 °C, reaction times shorter than 2 hours also yield amorphous IrO_x, while longer reaction times only slightly increase the nanorod length.

Therefore, within our current synthesis system, varying the reaction parameters does not enable control over the tunnel mouth size or the formation of shorter nanorods. We have added further discussion and supporting data in the revised manuscript and Supplementary Information (see page 15: “The effect of calcination time is examined...” and Supplementary Figs. 22-24).

Comment 4: In the experimental section, the acid treatment in 1M HClO₄ was essential for leaching out K⁺ ions. During this process, it seems that the IrO_x tunnel could undergo over-oxidation or dissolution, potentially leading to Ir loss in the catalyst. The author needs to explain these issues.

Response 4: Thanks. To address this concern, we conducted experiments to monitor possible Ir dissolution during the acid treatment (1 M HClO₄, 24 h). Inductively coupled plasma (ICP) analysis of the solution revealed that the Ir concentration was below the detection limit, indicating negligible Ir loss during acid treatment. These results demonstrate that the tunnel-structured IrO_x remains stable under acidic conditions.

Comment 5: In the OER mechanism, the oxidation state of Ir is a critical factor affecting both OER kinetics and stability. The authors should conduct additional XPS (X-ray photoemission spectroscopy) or XAS (X-ray absorption spectroscopy) analysis to examine the Ir oxidation states in the catalyst before and after electrochemical tests to support the OER performances.

Response 5: We thank the reviewer for this suggestion. We performed Ir L₃-edge X-ray absorption near-edge structure (XANES) analysis on T-IrO_x before and after electrochemical testing. Quantitative fitting of the white line intensity indicates that the Ir oxidation state in T-IrO_x is 3.2±0.2, while in T-IrO_x-OER, the oxidation state is 3.1 (Supplementary Fig. 33). This negligible change demonstrates that the Ir oxidation state remains nearly unchanged during OER, further supporting the excellent structural stability of T-IrO_x. We have added further discussion and supporting data in

the revised manuscript page 18: “Such remarkable structural stability could be attributed to...” and Supplementary Information (Supplementary Fig. 33).

Comment 6: The T-IrO_x-400 exhibited the highest OER activity compared to other T-IrO_x nanorod catalysts in Figure 5. The authors also calculated the intrinsic activity (j_Q) by normalizing the current with respect to the pseudocapacitive charge (Q_s). However, T-IrO_x-400, with the smallest aspect ratio, is expected to have the largest catalytic active area. Therefore, additional normalized current densities based on the electrochemically active surface area (ECSA) are required to accurately compare the intrinsic OER activities.

Response 6: Thanks. For iridium-based oxides with strong pseudocapacitive features, normalization by Q is widely adopted (*Nat. Catal.* **1**, 508-515 (2018); *J. Am. Chem. Soc.* **137**, 13031-13040 (2015)), and j_Q and j_{ECSA} are fundamentally equivalent metrics for evaluating intrinsic OER activity. At the reviewer’s suggestion, we have measured the ECSA for all T-IrO_x samples synthesized at different temperatures and normalized the current densities accordingly (j_{ECSA}). Both the ECSA and j_{ECSA} at 1.53 V follow the order: T-IrO_x-400 > T-IrO_x-500 > T-IrO_x-600 > T-IrO_x-700, which is fully consistent with the trends observed for Q_s and j_Q . This confirms our conclusions regarding the relationship between aspect ratio and intrinsic activity. Relevant data and discussion have been added to the revised manuscript (see page 16: “To further validate the intrinsic activity trend, we...”) and Supplementary Information (Supplementary Fig. 27).

Comment 7: According to various experimental and theoretical data presented in this study, the OER activities of the T-IrO_x catalyst appear to be strongly dependent on the tunnel mouth structure. However, these results fail to fully explain the advantages of the tunnel structure in T-IrO_x catalyst. Without addressing the synergistic effect between the tunnel mouths and tunnel walls, or the catalytic reactions occurring inside the tunnel, the tunnel structure does not seem to offer significant benefits compared to nanorod or nanoribbon structures. A more detailed and clear explanation of how the tunnel structure influences catalytic performance is required.

Response 7: Thanks! In the revised manuscript, we have provided a more detailed discussion on the tunnel structure-property relationships. The key points are as follows:

(i) The tunnel framework inherently favors the formation of 1D nanostructures. By properly tuning the nanorod length, we are able to maximize the exposure of highly active tunnel mouth sites while maintaining the advantages of 1D morphology—namely, the construction of efficient conductive networks and gas/liquid transport pathways within the catalyst-coated membrane (CCM). This structural balance simultaneously ensures the high CCM performance we observed.

(ii) The tunnel structure also helps stabilize Ir in a sub-4 oxidation state, which suppresses over-oxidation during OER and results in significantly lower Ir dissolution compared to rutile IrO_x, as confirmed by ICP-OES analysis (see page 4: “Besides, the iridium oxidation state in T-IrO_x is intrinsically stabilized at sub-4...”, page 18: “Such remarkable structural stability could be attributed to...” and Supplementary Fig. 32).

(iii) The tunnel architecture provides the structural foundation for the unique coordination environment at the tunnel mouths, which are identified as the key highly active sites through both theoretical calculations and experimental correlation. In contrast, rutile IrO_x nanowires and nanofibers exhibit much lower catalytic performance (*ACS Appl. Mater. Interfaces* **16**, 52179-52190 (2024); *New J. Chem.* **46**, 3716-3726 (2022)). In addition, rutile IrO_x nanorods were also synthesized

(Response Fig. 1), and the results show that both their apparent and intrinsic activities are inferior to those of T-IrO_x nanorods, further confirming the superior performance enabled by the tunnel structure.

Response Fig. 1 (a) XRD pattern of R-IrO_x nanorods. (b) TEM image of R-IrO_x nanorods. (c) iR -corrected OER polarization curves of R-IrO_x nanorods and T-IrO_x nanorods synthesized at different temperatures. (d) Pseudocapacitive charge-normalized OER activities (j_0) of R-IrO_x nanorods and T-IrO_x nanorods synthesized at different temperatures.

In summary, while the 1D nature of tunnel-structured IrO_x limits tunnel mouth exposure in long nanorods, we demonstrate that morphology optimization can reconcile this trade-off, enabling us to leverage the intrinsic advantages of the tunnel structure to achieve high performance in PEMWE.

Reviewer #2:

Comment: The manuscript reports the development of tunnel-structured IrO₂ catalysts for the oxygen evolution reaction in acidic medium.

I have been asked to assess the computational part, so I will not comment on the experimental achievements.

The first surprise comes in the abstract, where the authors state that the "tunnel mouths" are most active, but that nanorods are most active. One would have expected that plaquettes, rather than rods, should be optimal if the cross-section of the tunnel is most active compared to the outer-walls. Then, the introduction discusses the limitations of IrO₂ as stemming from 3 aspects: insufficient

active sites per surface area, charge-transfer issues and nanoparticle agglomeration.

The computational part of the study (and the resulting rationalisation for the observed performances) do not really come back to these limitations: the charge-transfer is discussed, but the computations do not yield valuable insight on this aspect. Other than that, the computational study claims higher intrinsic activity of the created "mouth" sites. However, according to the introduction, the intrinsic activity of IrO₂ is already good enough, so the computational study seems to miss the point.

Furthermore, not much is said to explain the supposedly significant difference between the T-IrO_x in the literature and the one presented herein: What is the critical difference that activates the current T-IrO_x compared to the ones studied before? This is critical, as the computational study suggest that R-IrO₂ is pretty inactive. Hence there is some contradiction to be dealt with.

The computational method is rather crude, in the sense that it does not include the explicit consideration of the electrochemical potential, nor explicit solvation. The former is key for discussing "protonation" (vs. hydrogenation; this is important in view of the ion-exchange) and the later (at least microsolvation) would be important to know the energy for water to enter into the tunnels.

In summary, the work seems carefully carried out, but overall coherence within the manuscript and the scientific hypothesis/reasoning requires significant improvements before the manuscript could be publishable anywhere.

Details:

- "fundamentally redefining design principles for iridium oxide catalysts": This is a too strong statement compared to the content of the manuscript.

- All the computed structures should be made available in the supplementary information or as a webarchive (typically on nomad-lab.eu). This is key for reproducibility.

- An OER overpotential of 0.5 V should not be qualified as "excellent".

Response: We thank the reviewer for the comments! Below, we address each of the points raised:

● **On the “nanorods vs. plaquettes” question in the abstract**

Our study identifies tunnel mouths as the primary catalytically active motifs. However, due to the intrinsic 1D crystal growth habit of tunnel-structured oxides, exposure of tunnel mouths is inherently limited. To date, all experimentally reported tunnel-type iridates form as nanowires or nanofibers as the consequence of this 1D crystal growth habit (*ACS Appl. Mater. Interfaces* **8**, 820-826 (2016)). Therefore, our study has focused on shortening the nanorods as much as possible to maximize the proportion of exposed tunnel mouths. While the synthesis of plaquette-like structures, as suggested by the reviewer, would indeed further increase tunnel mouth exposure, such morphologies are currently not accessible by available experimental methods.

● **Clarification on the scope of the computational study relative to device-level limitations**

We would like to clarify that the three limitations of IrO₂ mentioned in the Introduction, namely (i) insufficient active sites per surface area, (ii) charge-transfer issues, and (iii) nanoparticle agglomeration, refer specifically to the behavior of rutile-IrO₂ based membrane electrode assembly (MEA) of PEM electrolyzers. These are macroscopic or device-level limitations observed under practical conditions, rather than fundamental limitations of the IrO₂ electrocatalyst itself.

In a PEM electrolyzer, overall performance is governed by the combined contributions of

kinetic (η_{kinetic}), ohmic (η_{ohmic}), and transport ($\eta_{\text{transport}}$) overpotentials. η_{kinetic} is directly related to the intrinsic activity and utilization of the catalyst, η_{ohmic} is determined by the resistance of the catalyst layer, and $\eta_{\text{transport}}$ is primarily affected by the micro- and nanostructure that facilitates gas-liquid diffusion within the catalyst layer.

Currently, theoretical calculations can provide guidance on intrinsic catalytic activity, which correlates with η_{kinetic} . However, the catalyst layer structure in PEM electrolyzers is highly complex and is influenced by multiple factors (e.g., fabrication process), which are beyond the scope of current theoretical methods. Besides, the conductivity of catalyst layer is not determined solely by the conductivity of individual catalyst particles, but rather by the connectivity between particles within catalyst layer, which is another aspect that cannot be accurately predicted by theoretical calculation. Therefore, our computational efforts are focused on the properties directly related to intrinsic catalytic activity. Our experimental results further confirm that catalysts with higher theoretical activity also exhibit reduced η_{kinetic} in PEM electrolyzers, demonstrating a clear correlation between theoretical predictions and device performance.

- **On the performance gap between previously reported tunnel iridium oxides and T-IrO_x in this work**

We agree with the reviewer that a key question is why previously reported tunnel-structured IrO_x systems underperform, while the current version exhibits significantly improved activity. In fact, this contradiction was a central motivation for our study. In previous studies, tunnel iridium oxides were generally synthesized as long nanowires or nanofibers, mainly due to the high temperatures required in their synthetic protocols. In our work, we developed a synthesis approach that enables the formation of much shorter nanorods at lower temperatures, thereby significantly increasing the proportion of exposed tunnel mouths. This enhanced exposure of active sites directly leads to superior catalytic performance. Relevant discussions have been added to the revised manuscript (see page 16: “This result also explains why the unsatisfied activity of previously studied...”).

- **On the computational methodology and modeling scope**

In this study, we adopted the computational hydrogen electrode (CHE) model combined with standard DFT as a widely accepted and practical framework for evaluating relative catalytic trends among various iridium oxide configurations (*J. Phys. Chem. B* **108**, 17886-17892 (2004)). Over the past two decades, this approach has been extensively applied and validated for qualitative comparison and mechanistic hypothesis generation in OER catalyst design, with results showing good agreement with experimental observations (*Nat. Catal.* **3**, 516-525 (2020); *J. Am. Chem. Soc.*, **146**, 8915-8927(2024)).

With respect to ion exchange, the key process in our tunnel-structured iridium oxides under acidic conditions is the replacement of K⁺ ions within the tunnel by protons (H⁺) from the acidic solution, forming H_xIrO₂. The total number of electrons in the system remains unchanged throughout this process, and all calculations were carried out on charge-neutral models. This is a prototypical example of protonation (ion exchange) and does not involve electron transfer from an external circuit or the addition of neutral hydrogen atoms, which would be characteristic of hydrogenation. We employed chemical potential corrections within the CHE framework to model the protonation energetics, a widely used approach for predicting the thermodynamic tendencies of ion-exchange materials (*Science* **353**, 1011-1014 (2016); *Science* **383**, 1210-1215 (2024)).

- **On water insertion and inner site activity**

We performed calculations on water insertion thermodynamics across all investigated tunnel structures. For narrow tunnels (1×2 and $1 \times 2'$), water molecules are unable to enter the tunnel. For larger tunnels (2×2 , 2×3 , and 3×3), water molecules can be accommodated, but the thermodynamic driving force for water entry is small (0.02–0.03 eV/atom). We further evaluated the OER activity for inner wall active sites in the presence of water. Our results demonstrate that for more than 80% of inner active sites, the OER cycle cannot be completed due to the instability of key intermediates such as OOH^* ; for the remaining sites (~18%), the calculated overpotentials exceed 1.5 V, indicating that these inner active sites are theoretically inert. We have now added a detailed discussion of these results in the revised manuscript (see page 7: “In addition, we evaluated the accessibility of the tunnel interior to water...” and page 8: “The possibility of OER occurring at the inner active sites...”).

- **Clarification of language and data availability**

We have revised the wording throughout the manuscript to avoid overstatements and have adopted more appropriate descriptions. Additionally, all computed structures will be made available in the Supplementary Data to ensure reproducibility.

Reviewer #3:

Comment 1: The authors postulated through DFT calculations that tunnel mouth sites exhibit higher catalytic activity than tunnel wall sites in tunnel-structured IrO_x (T- IrO_x) materials. To experimentally validate this hypothesis, they synthesized a series of T- IrO_x -XXX nanorod samples by modulating synthesis temperatures to control the exposure ratio of tunnel mouth sites. The resultant T- IrO_x -400 sample applied in PEM electrolyzers achieve activity and outstanding long-term durability at large current density 2 A cm^{-2} . They proposed that the superior performance of T- IrO_x -400 was attributed to its shortened nanorod morphology, which purportedly maximizes the density of active tunnel mouth sites. However, a critical question arises regarding the potential confounding influence of structural defects. The relatively low synthesis temperature (400°C) may result in reduced crystallinity, as evidenced by the broadened diffraction peaks in the XRD patterns (Fig. 4b). This raises the possibility that the enhanced activity of T- IrO_x -400 could stem from defect-rich surfaces rather than the proposed tunnel mouth site dominance. The authors should provide more evidence to decouple the contributions of crystallographic defects from tunnel mouth site exposure (e.g., operando XPS for oxygen vacancy quantification, or operando EXAFS for local disorder assessment).

Overall, this study presents an innovative approach to probing the relationship between distinct tunnel structures and intrinsic activity. However, major revisions are required to address the aforementioned questions and several important issues.

Response 1: We thank the reviewer for their positive assessment of our work and for raising this important question regarding the possible influence of structural defects on the catalytic performance of T- IrO_x -400.

First, we would like to clarify that the broadened diffraction peaks observed in the XRD

patterns of T-IrO_x-400 (Fig. 4b) reflect the smaller nanorod size of this sample, rather than poor crystallinity. As shown in the Rietveld-refined XRD data (Supplementary Fig. 12) and the aberration-corrected TEM image (Fig. 4g), T-IrO_x-400 exhibits high crystallinity and a well-defined structure.

Nevertheless, we agree that structural defects could potentially contribute to catalytic performance, particularly at lower synthesis temperatures. To investigate this possibility, we performed additional characterizations. Electron paramagnetic resonance (EPR) spectroscopy was used to detect the presence of oxygen vacancies. The T-IrO_x-400 sample does not show any EPR signals attributable to unpaired electrons, indicating the absence of oxygen vacancies (Supplementary Fig. 29). Furthermore, EXAFS fitting analysis reveals that the Ir-O coordination number in T-IrO_x-400 is 6.3 ± 0.3 , consistent with a fully coordinated structure and confirming the absence of oxygen defects (Supplementary Table 6). We also considered the possibility of iridium defects. Aberration-corrected TEM images (Fig. 4g) reveal a well-ordered arrangement of Ir atoms, with no observable Ir defects. These results indicate that structural defects are not a significant factor in the enhanced catalytic activity of T-IrO_x-400. Additional data and relevant discussion have been included in the revised manuscript (see page 17: "In addition to aspect ratio, structural defects could potentially contribute to...") and Supplementary Information (Supplementary Figs. 12 and 29 and Table 6).

Comment 2: In the electrochemical LSV characterization (Figure 5a), T-IrO_x-400 shows enhanced intrinsic activity compared to other higher temperature-controlled counterparts. However, critical evaluation of operational durability remains incomplete. While the manuscript demonstrates promising stability for T-IrO_x-400 through chronopotentiometric testing, the absence of comparative stability data for other temperature-varied samples weakens the performance superiority of T-IrO_x-400. To enhance the comprehensive evaluation, the stability data for other temperature-controlled variants should be provided.

Response 2: We appreciate the reviewer's constructive comments. We have added chronopotentiometric stability data for T-IrO_x-500, T-IrO_x-600, and T-IrO_x-700 under identical conditions (10 mA cm⁻², 0.1 M HClO₄) in Supplementary Fig. 30. The results show that among these samples, T-IrO_x-400 exhibits the superior performance, maintaining better catalytic activity for nearly 1000 hours with a voltage increase of 3.0 μV h⁻¹. T-IrO_x-500 and T-IrO_x-600 exhibit comparable stability to T-IrO_x-400, whereas T-IrO_x-700 shows markedly inferior stability, with rapid voltage rise during operation (283 μV h⁻¹). This degradation can be attributed to the mechanical detachment of the T-IrO_x-700 nanorods from the electrode substrate, which is due to their excessive particle size (with nanorod lengths reaching 640 nm). The relevant discussion has been included in the revised main text (see page 17: "To evaluate their catalytic stability...").

Comment 3: The manuscript defines the "mouth area ratio" in Figure 5b as the ratio of the bipyramidal area to the total surface area of the nanorod. To improve reader comprehension, I recommend including a step-by-step calculation example using a representative nanorod to illustrate this concept more concretely.

Response 3: Thank you for the constructive suggestion. A detailed step-by-step calculation for the "mouth area ratio" is now provided in the Supporting Information (Supplementary Fig. 28).

Comment 4: The XRD patterns presented in Figure 4b and Figure S11 currently only indicate the

crystallographic planes corresponding to each diffraction peak. To enhance the structural characterization, I recommend identifying the crystal system and space group through comparison with standard PDF card. Or, explicit references could be provided to substantiate the phase identification.

Response 4: Thanks. The crystal system (monoclinic) and space group (I2/m) of T-IrO_x-400 have been included in the Rietveld refinement results, as shown in Supplementary Fig. 12 and Supplementary Table 5. This structural assignment is consistent with the standard PDF card (PDF# 85-2185), and has been added to the corresponding figure caption.

Comment 5: The Raman spectra shown in Supplementary Fig. 25 exhibit two characteristic peaks at around 550 and 720 cm⁻¹. To enhance the readers' understanding of the tunnel-structured IrO_x materials, the authors should provide detailed assignment of these two spectral features.

Response 5: We appreciate the reviewer's suggestion. The peaks at approximately 550 cm⁻¹ and 720 cm⁻¹ correspond to the bending vibration (E_g mode) and symmetric stretching vibration (A_{1g} mode) of the Ir-O bonds, respectively. These assignments have been indicated in revised Supplementary Fig. 35, and a detailed description has been added to the corresponding figure caption.

Comment 6: In the PEMWE section, the authors present an SEM image of R-IrO_x nanoparticles after prolonged operation in Supplementary Figure 36, giving the observed degradation in R-IrO_x morphology. To enable a meaningful comparison of catalyst durability, it would be valuable to include corresponding post-operation SEM of T-IrO_x after extended PEMWE operation.

Response 6: We appreciate the reviewer's suggestion. We have conducted SEM imaging of the T-IrO_x-400-based catalyst layer after PEMWE operation. The results show that the interwoven nanorod network structure is well preserved, with no observable agglomeration, collapse, or detachment. This demonstrates the superior structural integrity of the T-IrO_x-400-based catalyst layer after long-term operation. The corresponding SEM image and relevant discussion have been included in the revised manuscript (see page 24: "In contrast, the 1D morphology and interwoven stacking...") and Supplementary Information (Supplementary Fig. 48).

Reviewer's Comments and Our Responses:

Reviewer #1 (Remarks to the Author):

Comment 1: The authors have made significant revisions in response to the initial reviewer feedback. It is clear that several key points raised in the first round have been carefully considered and addressed. The additional experimental data and analyses have notably improved the clarity and robustness of the study. The study's core message, focusing on the spatially localized catalytic activity at the tunnel-mouth regions of IrO_x nanorods through morphological control, is now more clearly emphasized. This enhanced presentation is well supported by the newly added operando and structural characterization results. Addressing the following comments will further strengthen the mechanistic insights and overall contribution, enhancing the manuscript's suitability for publication in Nature Communications.

Response: We sincerely thank the reviewer for the positive evaluation of our revised manuscript. We are pleased to see that the reviewer recognizes the substantial improvements made in response to the initial comments. We have further revised the manuscript to address the remaining concerns, as detailed below, and believe these revisions enhance both the mechanistic insights and the overall contribution of our work.

Comment 2: Clarification of ECSA-normalized intrinsic activity (j_{ECSA}):

One key point raised in the initial review was the importance of normalizing catalytic activity to the electrochemically active surface area (j_{ECSA}), particularly in light of the substantial variation in nanorod aspect ratios and morphologies across the T-IrO_x series. While the authors continue to rely primarily on pseudocapacitive charge normalization (j_{Q}), the manuscript would benefit from explicitly reporting j_{ECSA} values and providing a direct comparison between j_{Q} and j_{ECSA} trends. Such a comparison would help determine whether the observed performance enhancements originate from genuine improvements in intrinsic activity or from increased surface accessibility. If experimental limitations prevent the inclusion of j_{ECSA} , a discussion of potential

discrepancies and their implications for interpreting the structure-activity relationship would still be valuable.

Response: We thank the reviewer for the insightful suggestion. In electrocatalysis, the double-layer capacitance (C_{dl}) and pseudocapacitive charge (Q) are two commonly used descriptors for quantifying the electrochemically active surface area (ECSA), which can be used to normalize catalytic activity and evaluate intrinsic activity. These two methods differ in principle and are suited to different material systems (*J. Electroanal. Chem.* 1992, 327, 353-376; *Chem. Soc. Rev.* 2019, 48, 2518-2534; *Chin. J. Catal.*, 2021, 42, 1054-1077).

The C_{dl} method is the most widely used approach for quantifying the ECSA, as it is applicable to most of electrocatalytic materials. Unless otherwise specified, the term ECSA typically refers to the value determined by the C_{dl} method, and j_{ECSA} denotes the corresponding intrinsic activity normalized by this surface area. This approach is based on the proportional relationship between C_{dl} and ECSA (i.e., $ECSA = C_{dl}/C_s$), where C_s is the specific capacitance per unit area, which depends on the electrode material, surface state, and electrolyte. However, for IrO_x systems, the accuracy of the C_{dl} -based method is limited by several factors: (i) IrO_x surfaces are often hydrated or covered by OH species under acidic OER conditions, which reduces conductivity and impairs complete double-layer charging; (ii) the presence of overlapping redox transitions makes it difficult to isolate the non-faradaic double-layer contribution from pseudocapacitive processes; and (iii) there lacks a universally accepted C_s value for IrO_x , introducing considerable uncertainty in ECSA estimation.

In contrast, IrO_x materials exhibit well-defined redox features in the OER-relevant potential range (~0.9-1.22 V vs. RHE), corresponding to the formation of active Ir-OH, Ir-O or Ir-OOH species. The pseudocapacitive charge method, which integrates the redox peak areas, provides a direct measure of the number of surface-active sites involved in the OER process. This method has proven more suitable for estimating active site density in noble metal oxides, such as IrO_x , and is widely employed in both three-electrode and membrane-electrode configurations. This consideration underpins our initial choice to normalize catalytic activity using the Q -based approach in this work.

We also agree the reviewer's constructive suggestion to complement the Q -based normalization with ECSA estimation derived from the C_{dl} method where appropriate. Given the high crystallinity and good conductivity of the IrO_x nanorods studied, the C_{dl} method is applicable when the potential window is carefully selected to avoid redox overlap. Accordingly, we have included C_{dl} -derived ECSA data (Supplementary Fig. 31) in the revised manuscript and compared it with the Q -based results. Both normalization approaches yield consistent activity trends (Q , ECSA, j_Q , and j_{ECSA} all follow the order: T- IrO_x -400 > T- IrO_x -500 > T- IrO_x -600 > T- IrO_x -700), further validating our conclusions regarding active site density, intrinsic activity, and the structure-activity relationship. Relevant data and a comprehensive discussion have been incorporated into the revised manuscript and Supplementary Information (pages 16-17 and Supplementary Fig. 31).

Comment 3: Kinetic analysis via Tafel slope comparison:

We appreciate the authors' detailed characterization of intrinsic catalytic activity and the comprehensive presentation of activity data. However, the manuscript currently lacks Tafel slope analysis, which is essential to deepen mechanistic understanding. Given the study's focus on morphological effects, comparing Tafel slopes across the T- IrO_x series and R- IrO_x samples would provide valuable insight into how tunnel-mouth exposure influences OER kinetics. This analysis could clarify whether tuning the nanorod aspect ratio affects the rate-determining step or overall reaction mechanism. Including Tafel slope data will complement the existing results by revealing changes in reaction kinetics, thereby strengthening the interpretation of morphology-dependent catalytic behavior.

Response: We thank the reviewer for this valuable suggestion. In the revised manuscript, we have added Tafel slope measurements (Supplementary Fig. 29) for T- IrO_x samples synthesized at different temperatures, as well as for R- IrO_x , and conducted a systematic analysis. The results show that Tafel slopes increase with decreasing tunnel mouth area ratio (T- IrO_x -400: 44 mV dec⁻¹, T- IrO_x -500: 48 mV dec⁻¹, T- IrO_x -600: 53 mV dec⁻¹, T- IrO_x -700: 57 mV dec⁻¹), while R- IrO_x exhibits a slope of 64 mV dec⁻¹. This

trend suggests that greater exposure of tunnel-mouth sites facilitates faster OER kinetics, consistent with the LSV results.

Moreover, the Tafel slope provides mechanistic insight into the rate-determining step (RDS) of the OER (*J. Electrochem. Soc.* 1967, 114, 592-593; *J. Electrochem. Soc.* 2013, 160, H142-H154; *Photoelectrochemical Solar Fuel Production* (Springer), 2016, Chapter 2, 41-104). The reaction can be deconstructed into the following elementary steps, where M denotes the active site and “ads” indicates the adsorbed intermediate:

The slopes of ~120, 60, 40, 15 mV dec⁻¹, is generally associated with step 1, steps 1 and 2, step 3, step 4, respectively, being rate-determining. In this work, The Tafel slopes of T-IrO_x series reflect the combined contributions of both tunnel mouth and wall regions. For samples with longer nanorods (e.g., T-IrO_x-700), the fraction of tunnel-mouth region is relatively low, and the Tafel slope remains close to 60 mV dec⁻¹, indicating that OH adsorption and reorganization (steps 1 and 2) likely dominate the kinetics. As the nanorod length decreases, the relative contribution of tunnel-mouth sites increases, and the Tafel slope gradually decreases toward 40 mV dec⁻¹. This trend suggests a greater kinetic contribution from the deprotonation step (step 3). Relevant data and a comprehensive discussion have been incorporated into the revised manuscript and Supplementary Information (page 16 and Supplementary Fig. 29).

Comment 4: Structural stability of nanorod tips and clarification of TEM image assignment:

Regarding structural stability, the revised manuscript more clearly emphasizes the long-term durability of T-IrO_x-400, including the newly added SEM image of the catalyst layer after prolonged PEMWE operation (Supplementary Fig. 48), which shows that the interwoven nanorod network remains intact with no observable collapse or detachment. This addition addresses earlier concerns about overall structural degradation. However, previously raised questions about localized deformation at the

nanorod tips—identified as key catalytic hotspots—still warrant further clarification. Supplementary Fig. 37 shows visible tip deformation after extended operation, contrasting with the pristine morphology in Supplementary Fig. 15. The authors should clarify whether these changes involve superficial rounding, amorphization, or facet reorganization, and whether such alterations affect the integrity or activity of the exposed active sites. Providing such clarification would help reconcile the apparent structural changes with the sustained catalytic performance. Additionally, the sample shown in Supplementary Fig. 15 should be explicitly identified, as the tip morphology closely resembles that of T-IrO_x-700 (Fig. 4f), which could cause confusion given the importance of tip structure in the study's design rationale.

Response: We apologize for the previous lack of clarity regarding sample labeling. Supplementary Fig. 15 corresponds to as-synthesized T-IrO_x-700, which is now clearly indicated in the revised figure caption.

Regarding the observed differences in tip morphology between Supplementary Fig. 15 (T-IrO_x-700) and Supplementary Fig. 37 (T-IrO_x-400-OER), we clarify that these differences arise from synthesis-temperature-dependent growth completeness, rather than from OER-induced degradation. Specifically, higher-temperature samples (e.g., T-IrO_x-700) tend to exhibit more well-defined and sharper tips due to more complete nanorod formation, whereas lower-temperature samples (e.g., T-IrO_x-400) display smoother and more rounded terminations. To further support this, we have now included systematic TEM comparisons of all T-IrO_x samples before and after OER testing (Supplementary Figs. 15-18 and 42-44), showing that these morphological features are already present prior to OER operation. Besides, the nanorod tips of T-IrO_x series exhibit minimal morphological changes and retain high crystallinity after long-term operation. No signs of superficial rounding, amorphization, or facet reorganization are observed, confirming their excellent structural stability under acidic OER conditions. All supporting data have been included in the revised manuscript and Supplementary Information (pages 13-14 and 19 and Supplementary Figs. 41-44).

Although the tips of T-IrO_x nanorods display some morphological differences, this does not significantly impact their catalytic function. Theoretical results indicate that

all tunnel-mouth regions, whether sharp or rounded, exhibit similarly high intrinsic activity. Geometric modeling further shows that the tip shape introduces only minor changes to the calculated mouth area ratio. Thus, these variations do not alter our conclusion: spatial activity heterogeneity in T-IrO_x governs its performance, with increased tunnel-mouth exposure leading to higher catalytic activity.

Comment 5: Additional literature context for comparison of design strategies:

To provide further context and mechanistic perspective, the authors might consider referencing the recent study published in Nature Communications, titled “Atomic-level Ru-Ir mixing in rutile-type (RuIr)O₂ for efficient and durable oxygen evolution catalysis” (Nat. Commun. 16, 579, 2025). This work also investigates nanorod morphology, active site distribution, and catalyst stability under acidic OER conditions. Drawing a brief comparison to this study, e.g., how the present work emphasizes spatial engineering of tunnel mouths rather than atomic-level compositional tuning, would provide useful perspective and help position the current manuscript within the broader literature landscape.

Response: We thank the reviewer for this suggestion. In the revised manuscript, we have included a brief comparison with the referenced study (Nat. Commun. 2025, 16, 579), which utilizes atomic-scale Ru-Ir mixing to modulate the electronic structure and performance of rutile-type Ir-based nanorods. In contrast, our work focuses on spatial engineering of crystallographically distinct sites, specifically tunnel mouths, through morphology control in a pure IrO_x system. These two strategies represent complementary design principles: atomic-level compositional tuning versus nanoscale structural modulation, both targeting improved acidic OER performance. This comparison has also inspired new perspectives for future development of tunnel-structured electrocatalysts (page 21).

Reviewer #2 (Remarks to the Author):

The authors have performed a reasonably careful revision, but a few issues remain:

- The recent literature is insufficiently cited, e.g., doi/10.1002/sml.202412237 and, for the modelling part, DOI10.1039/d2ee00158f and 10.1002/wcms.1499, emphasizing the need to go beyond CHE in order to avoid over-simplified conclusions.

- The introduction/motivation of the study is still partially at odds with the final claims/conclusions: T-IrO₂ has “technical” advantages that could address the challenges stated on the top of the second page of the introduction. However, there seems to be no need for higher activity compared to R-IrO₂: Fig 3c reveals that 6-7 active sites on the tunnel walls should be as active as R-IrO₂, so that the poor performance of “standard” (high aspect ratio) remains a bit surprising. Furthermore, it should be clarified that the overpotential of R-IrO₂ is not the main issue and it is just a lucky coincidence that the mouths are a bit more active (Fig 6c), but it is not responsible for the main importance of the current study. Along the same lines, it might be useful to rephrase “with short nanorods achieving optimal balance between active site exposure and electron/mass transport efficiency”: According to my understanding, the authors have no means to claim “optimal balance”: The boundary (the shortest nanorods that are achievable) are the most active, but they have no idea if even shorter nanorods would not perform even better. To summarize this somewhat vague point, I would like to invite the authors to tune down the claims and focus on the main message: shorter rods are more active and still stable. This applies to the main text and the first sentence of the “Discussion”.

After these rather minor modifications, I expect the manuscript to be publishable.

Response: We thank the reviewer for the constructive feedback and helpful suggestions. In the revised manuscript, we have carefully addressed the remaining concerns, as outlined below:

(i) Citation of recent literature and comment on the CHE model

We thank the reviewer for highlighting these important references and have now cited all three works in the revised manuscript.

The study by Moss et al. (Small, 2025, 21, 2412237) offers a valuable example of how nanostructure engineering of nanocrystalline IrO₂ can enhance both activity and durability. Our work aligns with this strategy, where structural morphology is likewise leveraged to modulate surface accessibility and catalytic performance. We have added a brief discussion of this parallel in the revised manuscript to better contextualize our approach.

For the theoretical part, the works by Binninger et al. (*Energy Environ. Sci.*, 2022, 15, 2519-2528) and Abidi et al. (*WIREs Comput. Mol. Sci.*, 2021, 11, e1499) provide valuable insight into the evolution of computational methodologies beyond the conventional computational hydrogen electrode (CHE) model. While such beyond-CHE approaches are crucial for capturing interfacial complexities such as solvation and electric fields, it is important to note that CHE remains the most widely adopted and practical framework for comparative evaluation of OER activity across similar surfaces. In our case, the trends predicted by CHE-based calculations show agreement with experimental observations, thereby reinforcing the reliability of our approaches. To provide readers with a more comprehensive view, we have added a brief statement in the revised manuscript discussing the scope and limitations of the CHE model, along with references to the suggested works (page 36). The reviewer's comment also highlights a meaningful future direction—the development of more comprehensive theoretical descriptions for electrocatalytic systems.

(ii) Revising the expression “optimal balance”

At present, experimental limitations prevent us from synthesizing T-IrO_x nanorods with even shorter lengths, which constrains our ability to identify a true optimal aspect ratio. Nevertheless, the current results establish a clear spatial engineering strategy through morphology control. This approach allows us to confirm that tunnel-mouth sites are catalytically dominant, and that shorter T-IrO_x nanorods not only demonstrate higher intrinsic activity but also offer *technical advantages*, both of which contribute to their improved performance in PEM water electrolyzers. To reflect this more accurately, we have revised the manuscript to remove the implication of having identified an “*optimal balance*”, and to adopt more appropriate expression.

(iii) Explaining the low apparent activity of high-aspect-ratio T-IrO_x

As for the relatively *poor performance of “standard” (high-aspect-ratio) T-IrO_x*, we note that although DFT results (Fig. 3c) indicate comparable theoretical activity between tunnel-wall sites of T-IrO_x and sites of rutile IrO₂, the large particle size and limited active-site exposure in these nanorods significantly reduce their surface accessibility. In contrast, commercial R-IrO_x typically comprises nanoparticles (<5 nm), offering much greater accessible surface area and thereby enhanced apparent activity. This discrepancy helps explain the lower apparent activity of *“standard” T-IrO_x* relative to R-IrO_x, despite their comparable theoretical activity.